# Oligomerization of p62 allows for selection of ubiquitinated cargo and isolation membrane during selective autophagy

Bettina Wurzer[†], Gabriele Zaffagnini[†], Dorotea Fracchiolla, Eleonora Turco, Christine Abert, Julia Romanov, Sascha Martens*

Max F. Perutz Laboratories, University of Vienna, Vienna Biocenter, Vienna, Austria

**Abstract** Autophagy is a major pathway for the clearance of harmful material from the cytoplasm. During autophagy, cytoplasmic material is delivered into the lysosomal system by organelles called autophagosomes. Autophagosomes form in a de novo manner and, in the course of their formation, isolate cargo material from the rest of the cytoplasm. Cargo specificity is conferred by autophagic cargo receptors that selectively link the cargo to the autophagosomal membrane decorated with ATG8 family proteins such as LC3B. Here we show that the human cargo receptor p62/SQSTM-1 employs oligomerization to stabilize its interaction with LC3B and linear ubiquitin when they are clustered on surfaces. Thus, oligomerization enables p62 to simultaneously select for the isolation membrane and the ubiquitinated cargo. We further show in a fully reconstituted system that the interaction of p62 with ubiquitin and LC3B is sufficient to bend the membrane around the cargo.

*For correspondence: sascha.martens@univie.ac.at

[†]These authors contributed equally to this work

**Competing interests:** The authors declare that no competing interests exist.

## Introduction

Cellular homeostasis and quality control require degradation of potentially harmful cytoplasmic material. The lysosomal system mediates degradation of large and bulky substances that cannot be degraded by other means, for example, the proteasome. A major pathway for the degradation of cytoplasmic material is macroautophagy (hereafter autophagy) (*De Duve and Wattiaux, 1966*). During autophagy, double membrane-bound organelles called autophagosomes are formed that, upon fusion with the lysosomal system, deliver cytoplasmic cargo material for degradation (*Kraft and Martens, 2012*). Autophagosomes form in a de novo manner. Initially, small-membrane structures called isolation membranes or phagophores are observed, which gradually enclose cargo material as they grow. Upon closure of the isolation membranes, autophagosomes are formed within which the cargo is isolated from the rest of the cytoplasm. Subsequently, the autophagosomes fuse with the endolysosomal system where the inner membrane and the cargo are eventually degraded (*Kraft and Martens, 2012*).

It has become clear that autophagy can be highly selective with regard to the cargo that is enclosed and degraded (*Rogov et al., 2014*). Among the many cargos are aggregated proteins (*Bjørkøy et al., 2005*; *Kirkin et al., 2009a*; *Komatsu et al., 2007*; *Pankiv et al., 2007*; *Szeto et al., 2006*), damaged mitochondria (*Geisler et al., 2010*; *Kanki et al., 2009*; *Narendra et al., 2008*; *Novak et al., 2010*; *Okamoto et al., 2009*), intracellular pathogens (*Gutierrez et al., 2004*; *Nakagawa et al., 2004*; *Thurston et al., 2009*; *Yoshikawa et al., 2009*; *Zheng et al., 2009*), surplus peroxisomes (*Farré et al., 2008*; *Hutchins et al., 1999*; *Iwata et al., 2006*), and ferritin (*Dowdle et al., 2014*; *Kishi-Itakura et al., 2014*; *Mancias et al., 2014*). Consequently, dysfunctional

**eLife digest** Cells use a process called autophagy to destroy damaged proteins and other harmful materials. During autophagy, the harmful materials become enclosed in a compartment called an autophagosome, which seals it off from the rest of the cell. The autophagosome – which is made of a double-layered membrane – then fuses with another compartment called a lysosome, which contains enzymes capable of breaking down the harmful material. Autophagy helps to keep cells healthy and defects in this process can contribute to cancer, neurodegeneration and other serious diseases in humans.

It is important that cells are able to correctly select harmful materials for autophagy. A protein called p62 is able to specifically bind to damaged proteins that have been covered with a molecule called ubiquitin. It also interacts with proteins of the ATG8 family that are found on the surface of the developing autophagosome. In this way, p62 operates as an adaptor that brings damaged proteins into contact with autophagosomes in preparation for autophagy. However, since a single p62 molecule can only interact with one ubiquitin molecule and one ATG8 protein, it is not clear how it is able to specifically select only the proteins that carry many ubiquitin tags.

Individual p62 units can interact with each other to form a group called an oligomer. Wurzer, Zaffagnini et al. use a combination of biochemical and cell biological techniques to study these oligomers in human cells. The experiments show that the oligomers are able to bind to many ubiquitin tags on a single structure. This enables a p62 oligomer to form a stronger connection to the damaged protein than a single p62 unit can. At the same time, the oligomer can interact with many ATG8 proteins, which tend to be found in clusters only on the surface of the autophagosome.

Previous studies have shown that when an autophagosome starts to form, its membrane expands and curves around the cargo. Wurzer, Zaffagnini et al. observed that the binding of p62 oligomers to ATG8 proteins and damaged structures with ubiquitin tags, drive the bending of the membrane around these structures. Wurzer, Zaffagnini et al.'s findings reveal how the formation of oligomers allows p62 to specifically select and target damaged proteins that are covered with many ubiquitin tags for destruction, while sparing other materials. Other proteins that are closely related to p62 also help to select cell materials for autophagy, so a future challenge is to find out whether these proteins work in a similar way.

autophagy results in several pathological conditions such as neurodegeneration, cancer, and uncontrolled infection (*Levine and Kroemer, 2008*; *Mizushima and Komatsu, 2011*; *Levine et al., 2008*).

The selectivity of autophagic processes is conferred by autophagic cargo receptors that bind the cargo and link it to the isolation membrane (*Johansen and Lamark, 2011*). The isolation membrane is specifically recognized by the cargo receptors due to its modification with proteins of the ATG8 family (*Kabeya et al., 2000*). Yeast Atg8 and its homologues are ubiquitin-like proteins that become conjugated to the headgroup of the membrane lipid phosphatidylethanolamine (*Ichimura et al., 2000*). This unusual modification renders the soluble ATG8 proteins membrane-bound and serves as an identifier for the isolation membrane (*Ichimura et al., 2000*).

The first autophagic cargo receptor identified was the *Saccharomyces cerevisiae* Atg19 protein (*Leber et al., 2001*; *Scott et al., 2001*). Atg19 acts during the transport of the oligomeric prApe1 peptidase and other cargos into the vacuole (*Hutchins and Klionsky, 2001*; *Scott et al., 2001*; *Suzuki et al., 2011*; *Yuga et al., 2011*). Within the vacuole, prApe1 becomes activated and fulfills its enzymatic function. Under basal, nutrient-rich conditions, the prApe1 oligomers are constitutively transported into the vacuole by the cytoplasm-to-vacuole transport (Cvt) pathway (*Klionsky et al., 1992*), in which small double membrane-bound vesicles, called Cvt vesicles, tightly enclose the prApe1 cargo (*Baba et al., 1997*). The formation of these Cvt vesicles depends on the core autophagic machinery (*Harding et al., 1995*) and it is mechanistically analogous to the formation of selective autophagosomes in complex eukaryotes, including mammals (*Lynch-Day and Klionsky, 2010*). The Atg19 receptor binds directly and strongly to the prApe1 cargo (*Morales Quinones et al., 2012*; *Sawa-Makarska et al., 2014*; *Scott et al., 2001*; *Shintani et al., 2002*). In addition, it contains multiple Atg8 binding sites (*Noda et al., 2008*; *Sawa-Makarska et al., 2014*; *Shintani et al., 2002*).

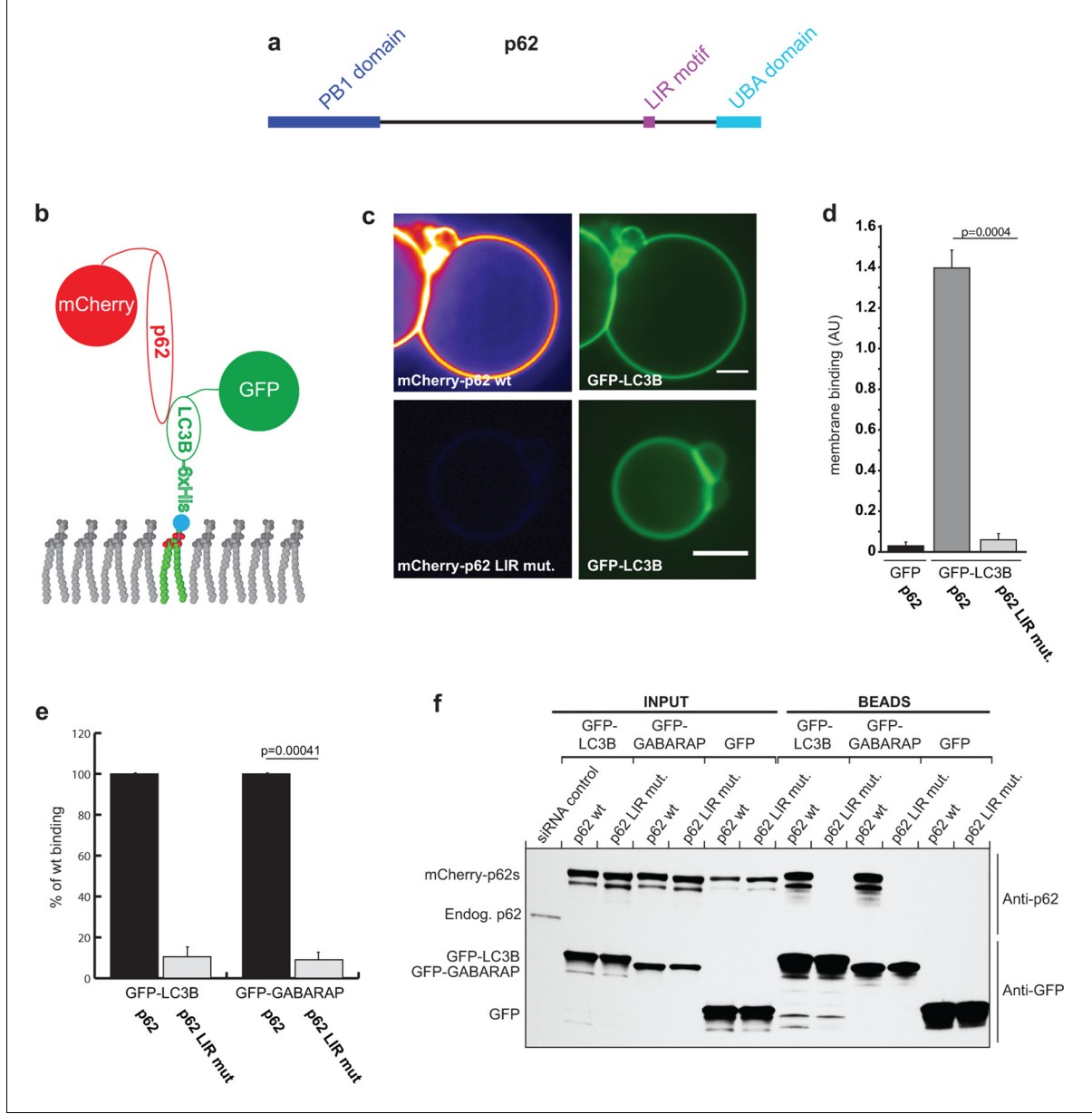

**Figure 1.** p62 has only one detectable LC3B/GABARAP-interaction motif. (**A**) Schematic representation of the p62 domain architecture. (**B**) Scheme showing the experimental set-up of the assay. GFP-LC3B-6xHis or GFP-6xHis were tethered to giant unilamellar vesicles (GUVs) via nickel-lipids incorporated into the membranes. Recombinant wild-type or LIR mutant mCherry-p62 were added and their membrane recruitment was determined. (**C**) Representative images of GUVs incubated with GFP-LC3B-6xHis and mCherry-p62 wild-type or LIR mutant. The mCherry signal is shown in false color (ImageJ: fire). Scale bars, 5 μm. (**D**) Quantification of mCherry-p62 wild-type or LIR mutant membrane recruitment. Averages and SD from three independent experiments are shown. The p-value was determined using a two-tailed unpaired Student's t-test. (**E**) Quantification of mCherry-p62 wild-type (black bars) or LIR mutant (gray bars) recruitment to GFP-LC3B-6xHis or GFP-GABARAP-6xHis coated GUVs. Data are normalized to the wild-type p62 binding intensity. The error bars are derived from three independent experiments. The p-value was determined using a two-tailed unpaired Student's t-test. (**F**) Anti-GFP and anti-p62 western blots of input (8%) and bead (50%) fractions of a GFP-TRAP pull-down of HeLa cell lysates co-expressing GFP-LC3B, GFP-GABARAP, or GFP (control) and wild-type or LIR mutant mCherry-p62. The endogenous p62 was silenced by siRNA. The experiment was conducted twice. (**D**) Total GUVs counted per condition: GFP-LC3B-6xHis + mCherry-p62 wild-type = 163; GFP-LC3B-6xHis + mCherry-p62 LIR mutant = 152; GFP-6xHis + mCherry-p62 wild type = 254. (**E**) Total GUVs counted per condition: GFP-LC3B-6xHis + mCherry-p62 wild type = 636; GFP-LC3B-6xHis + mCherry-p62 LIR mutant = 642; GFP-GABARAP-6xHis + mCherry-p62 wild type = 336; GFP-GABARAP-6xHis + mCherry-p62 LIR mutant = 300.

These two properties enable Atg19 to bend the membrane tightly around the cargo and thereby to exclude non-cargo material from the Cvt vesicles (*Baba et al., 1997*; *Sawa-Makarska et al., 2014*).

Mammals have multiple cargo receptors that mediate the autophagic degradation of cytoplasmic material (*Johansen and Lamark, 2011*). While some mammalian cargo receptors such as NCOA4 directly recognize their cargo (*Dowdle et al., 2014*; *Mancias et al., 2014*), many mammalian cargo receptors including p62/SQSTM-1, NBR1, Optineurin, NDP52, and Tollip recognize the cargo material due to its modification with ubiquitin (*Bjørkøy et al., 2005*; *Kirkin et al., 2009*; *Kirkin et al., 2009*; *Lu et al., 2014*; *Rogov et al., 2014*; *Thurston et al., 2009*; *Wild et al., 2011*). p62 is a multidomain protein and contains, among other domains, an N-terminal PB1 domain, a LIR motif mediating the interaction with ATG8 family proteins and a C-terminal UBA domain that binds ubiquitin (*Figure 2*) (*Johansen and Lamark, 2011*; *Vadlamudi et al., 1996*). The affinity of the UBA domain for ubiquitin is very low (*Long et al., 2008*; *Long et al., 2010*) but can be increased by phosphorylation on serine 403 (*Matsumoto et al., 2011*). The N-terminal PB1 domain mediates interaction with several other proteins as well as homo-oligomerization (*Lamark et al., 2003*). Recently, it was shown by cryo-electron microscopy that in vitro p62 forms large helical structures in a PB1-dependent manner (*Ciuffa et al., 2015*).

Mutations in the PB1 domain that interfere with its ability to oligomerize inhibit the recruitment of p62 to the autophagosome formation site (*Itakura and Mizushima, 2011*). Moreover, deletion of the PB1 domain or oligomerization-inhibiting mutations decrease the interaction with both LC3B and ubiquitin in pull-down assays, suggesting that oligomerization may increase the interaction with these binding partners.

Here we show in a variety of in vitro and in vivo systems that the oligomerization of p62 generates high-avidity interactions with ubiquitin and LC3B-coated surfaces, which allows p62 to select for cargo material and the isolation membrane. In particular, we show that oligomerization does not increase the affinity of each individual binding site but reduces the off-rate of the oligomeric unit from ubiquitin and LC3B-coated surfaces, respectively. We further show in a reconstituted system that the concurrent interaction of p62 with ubiquitin and LC3B is sufficient to drive the close apposition of the membrane and the cargo.

## Results

### Interaction of p62 with ATG8 family proteins depends on a single LIR motif

The *S. cerevisiae* Atg19 cargo receptor contains multiple low-affinity Atg8 binding sites that enable it to selectively and tightly bind to membrane-localized Atg8 (*Sawa-Makarska et al., 2014*). We asked whether this feature is conserved and turned our attention to p62, which is a major cargo receptor in mammals, including humans. Only a single LIR motif has been identified in p62 (*Ichimura et al., 2008*; *Pankiv et al., 2007*), but there was the possibility that low-affinity-binding sites for ATG8 family proteins such as LC3B and GABARAP were not detected in classical pull-down assays since they fail to detect interactions with high off-rates. We, therefore, used a more sensitive assay to find potential p62–LC3B interaction sites that are independent of the known LIR motif. To this end, we attached GFP-labeled LC3B or GABARAP to the membrane of giant unilamellar vesicles (GUVs). Recombinant mCherry-p62 was added to the GFP-LC3B and GFP-GABARAP-coated GUVs and the recruitment of mCherry-p62 was followed by spinning disk microscopy (*Figure 2*). mCherry-p62 was robustly recruited to GFP-LC3B and GFP-GABARAP but not to GFP-coated GUVs. Upon simultaneous mutation of D335, D336, D337, and W338 to A in the LIR motif of p62 (*Ichimura et al., 2008*; *Pankiv et al., 2007*), the recruitment of the protein to the GFP-LC3B and GFP-GABARAP-coated GUVs was completely abolished (*Figure 2*), strongly suggesting that p62 has only one functional LC3B/GABARAP interaction site. We will refer to this mutant as the LIR mutant.

We corroborated these results in GFP-TRAP experiments using HeLa cell lysates (*Figure 2*), where the interaction of p62 with LC3B and GABARAP totally depended on its LIR motif.

### Oligomerization of p62 promotes the interaction with LC3B

The N-terminal PB1 domain of p62 mediates oligomerization (*Ciuffa et al., 2015*; *Lamark et al., 2003*). Within the p62 oligomers, LIR motifs are clustered, similar to the occurrence of multiple Atg8

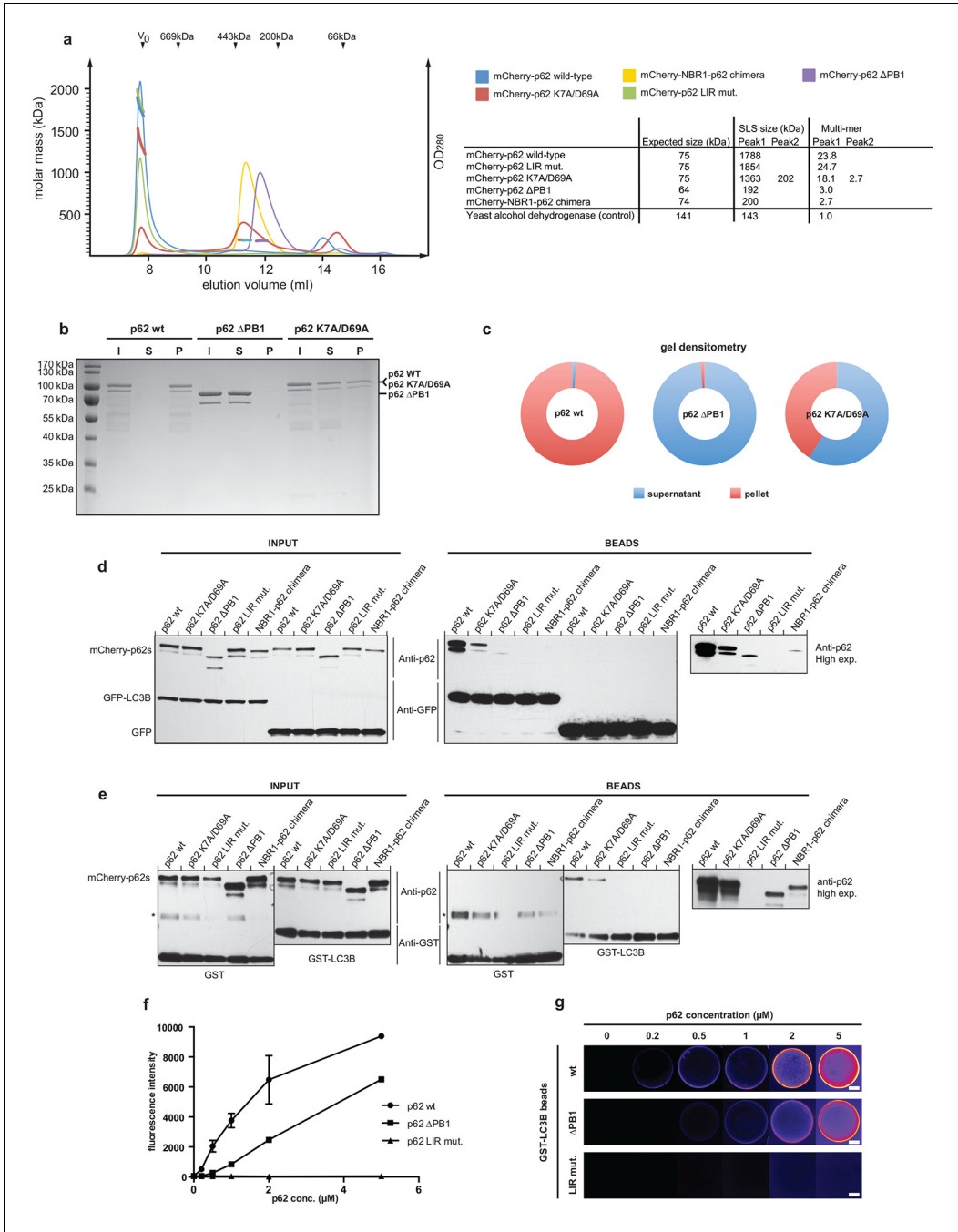

**Figure 2.** Oligomerization of p62 stabilizes binding to LC3B-coated surfaces. (**A**) Size exclusion chromatography (SEC) and static light scattering (SLS) analysis of recombinant wild-type mCherry-p62, the LIR mutant and the oligomerization mutants (K7A/D69A, delta PB1, and NBR1-p62 chimera). The left *Y*-axis indicates the molecular weight of the protein as determined by SLS. The average sizes of the indicated peak areas obtained by SLS are shown in the table. See *Figure 2* for gel. (**B**) Coomassie-stained gel showing a p62 sedimentation assay of recombinant mCherry-p62 wild-type, delta PB1, and K7A/D69A mutants. For each p62 variant input, supernatant and pellet fractions were loaded. (**C**) Quantification of the p62 sedimentation assay shown in (**B**). Amounts of p62 in the supernatant (blue) and pellet (red) are represented as fractions of the input. (**D**) Anti-GFP and anti-p62 western blot of input (8%) and bead (50%) fractions of a GFP-TRAP affinity purification of HeLa cell lysates co-expressing GFP-LC3B or GFP (control) and the mCherry-p62 variants. The endogenous p62 was silenced by siRNA treatment (*Figure 2—figure supplement 2*). A representative blot of four independent replicates is shown. (**E**) Anti-GST and anti-p62 western blot analysis of input (8%) and bead (16%) fractions of a pull-down experiment using GST-LC3B or GST (control) as bait and purified mCherry-p62 variants as prey. A representative blot of three independent replicates is shown. Asterisks denote dimeric GST. (**F**) Quantification of steady-state binding intensities of increasing concentration of wild-type, delta PB1, or the LIR mutant mCherry-p62 on GST-LC3B-coated beads. The average fluorescence intensity on the beads is plotted against the p62 concentration.

*Figure 2. continued on next page*

*Figure 2. Continued*

Averages and SD of three independent experiments are shown. (**G**) Representative images of the experiment shown in (**F**). The mCherry signal is shown in false color (ImageJ: fire). (**F**) Total beads quantified: wild-type 0.2 μM = 187 - 0.5 μM = 198 - 1 μM = 180 - 2 μM = 175 - 5 μM = 73; p62 delta PB1 0.2 μM = 133 - 0.5 μM = 163 - 1 μM = 179 - 2 μM = 176 - 5 μM = 58; p62 LIR mutant 0.2 μM = 74 – 0.5 μM = 84 – 1 μM = 75 – 2 μM = 85 – 5 μM = 75.

The following figure supplements are available for Figure 2:

**Figure supplement 1.** (**A**) Coomassie-stained gel showing the peak fractions of wild-type mCherry-p62 and the K7A/D69A mutant after the size exclusion chromatography (SEC)/static light scattering (SLS) runs.

**Figure supplement 2.** Western blot of samples shown in *Figure 2D* showing efficient siRNA-mediated silencing of endogenous p62 in mCherry-p62 co-transfected cells.

**Figure supplement 3.** Relative fluorescence intensity plot of data shown in *Figure 2F*.

binding sites in the Atg19 cargo receptor (*Sawa-Makarska et al., 2014*). Indeed, the PB1 domain was shown to enhance LC3B binding in pull-down experiments (*Bjørkøy et al., 2005*). To directly test whether the strength of the p62–LC3B interaction correlates with the ability of p62 to oligomerize, we recombinantly expressed and purified several oligomerization mutants of p62. The attachment of mCherry to the N-terminus of p62 considerably increased the yield of soluble protein. In order to determine the oligomerization state of our mCherry-p62 variants, we conducted size exclusion chromatography (SEC) runs coupled to static light scattering (SLS) (*Figure 2A* and *Figure 2— figure supplement 1*). This allowed us to determine the molecular mass of the p62 variants independently of their shape. The wild-type and LIR mutant proteins eluted in the exclusion volumes ($V_0$) of the column. SLS showed that the protein in the $V_0$ was composed of oligomeric particles of on average 24 molecules. Deletion of the PB1 domain resulted in a complete shift of the protein from the $V_0$ to lower molecular weight fractions. Interestingly, SLS showed that p62 delta PB1 is a trimer. The structural basis for the trimeric form is currently unknown.

Similarly, when we exchanged the PB1 domain of p62 for the non-oligomerizing PB1 domain of NBR1 (*Lamark et al., 2003*) the protein became trimeric. We will refer to this mutant as the NBR1-p62 chimera. Interestingly, introduction of the oligomerization-interfering K7A/D69A double mutation (*Lamark et al., 2003*) into the PB1 domain of p62 resulted in an intermediate behavior between the two extremes with a small fraction of the protein eluting in the $V_0$ and another peak representing the trimeric species (*Figure 2A* and *Figure 2—figure supplement 1*). To confirm this result, we tested purified wild-type mCherry-p62, p62 delta PB1, and the K7A/D69A mutant in a p62 sedimentation assay (*Ciuffa et al., 2015*). Consistent with the SLS results, the wild-type protein nearly completely sedimented into the pellet, while the delta PB1 mutant remained in the supernatant. Interestingly, the K7A/D69A mutant partitioned into both fractions (*Figure 2B,C*).

To analyze whether the ability of p62 to oligomerize correlates with the strength of its interaction with LC3B, we performed GFP-TRAP experiments using cell lysates of HeLa cells co-transfected with siRNA-resistant versions of the mCherry-p62 variants and GFP-LC3B (*Figure 2D*). The endogenous p62 was silenced by siRNA (*Figure 2—figure supplement 2*). Indeed, there was a strong correlation between the ability of p62 to oligomerize and its presence in the bead fraction (*Figure 2D*). While the wild-type protein showed the most robust interaction with LC3B, the interaction of the K7A/D69A double mutant (*Lamark et al., 2003*) was reduced but still readily detectable. The interaction of delta PB1 p62 and the NBR1-p62 chimera became detectable only after long exposure of the blots.

Next, we tested the different purified recombinant mCherry-p62 variants in pull-down assays using GST-LC3B as bait (*Figure 2E*). Similarly to what we observed in the GFP-TRAP experiments, the ability of the different p62 variants to co-pellet with GST-LC3B correlated strongly with their oligomeric state, suggesting that oligomerization of p62 directly affects its binding to LC3B.

Pull-down assays favor interactions with off-rates low enough to resist washing. Therefore, one possible interpretation of these results is that the oligomeric wild-type p62 has a lower off-rate from LC3B clustered on a surface than the non-oligomerizing mutants. However, it is also possible that in p62 oligomers some monomers are simply piggybacked without actively contributing to the

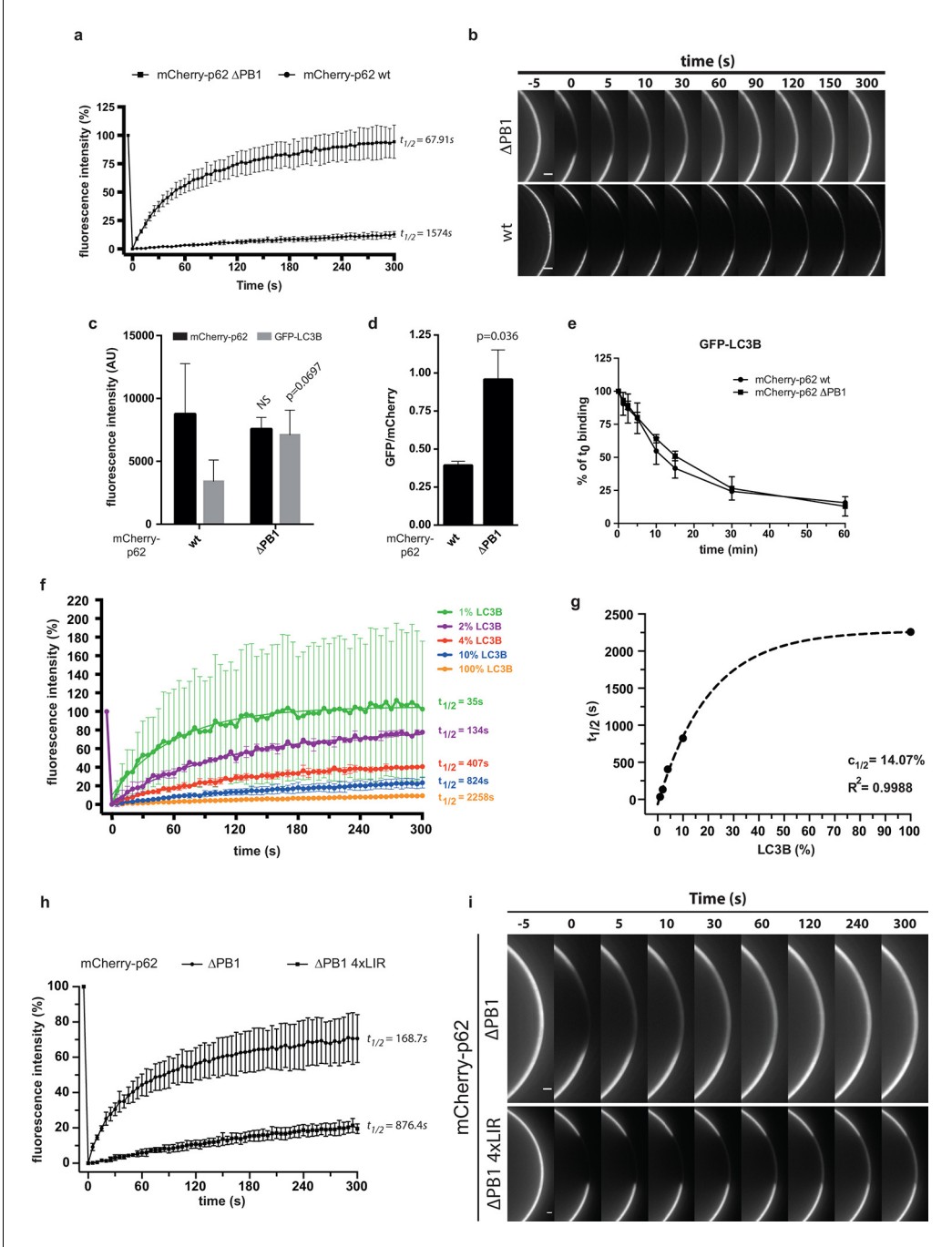

**Figure 3.** Oligomerization of p62 renders binding to concentrated LC3B irreversible. (**A**) Fluorescence recovery after photo-bleaching (FRAP) curve of the indicated mCherry-p62 variants on GST-LC3B-coated beads. Averages and SD of at least three independent curves are shown. (**B**) Representative pictures for the data shown in (**A**). Scale bar 5 µm (**C,D**). Quantification of steady-state binding of indicated mCherry-p62 variants to RFP-TRAP beads and of subsequent GFP-LC3B recruitment to these beads. Absolute fluorescence intensities are shown in (**C**). A plot of GFP/mCherry ratio is shown in (**D**). Averages and SD of three independent replicates are shown. Indicated p-values were calculated with a two-tailed unpaired Student's *t*-test. (**E**) Quantification of decay of GFP-LC3B fluorescence from RFP-TRAP beads coated with indicated mCherry-p62 variants. Averages and SD of two independent replicates are shown. (**F**) Fluorescence recovery (FRAP) curves of wild-type mCherry-p62 recruited to glutathione beads coated with decreasing amounts of GST-LC3B. Averages and SD of four independent curves per sample are shown. (**G**) Plot of extrapolated recovery half-times ($t_{1/2}$) from (**F**) against the respective LC3B concentration on the beads. Data points were fitted to a mono-exponential equation. Robustness of the fit ($R^2$)

*Figure 3. continued on next page*

*Figure 3. Continued*

and the extrapolated half-maximal LC3B concentration ($c_{1/2}$) are indicated. (H) FRAP curves of the indicated p62 variants on GST-LC3B coated beads. Averages and SD of four independent curves are shown. (I) Representative pictures for the graph shown in (H). Scale bar 20 µm. (C, D) Total beads quantified: wild type = 101, delta PB1 = 162. (E) Total beads quantified: wild type = 78, delta PB1 = 71. (Figure supplement 1) Total beads quantified: wild type = 98, delta PB1 = 133. (Figure supplement 2) Total beads quantified: wild type = 45, delta PB1 = 49. (Figure supplement 4) Total beads quantified per condition. Wild type: 0% LC3B = 150; 1% LC3B = 141; 2% LC3B = 130; 4% LC3B = 92; 10% LC3B = 119; 50% LC3B = 92; 100% LC3B = 132. delta PB1: 0% LC3B = 82; 1% LC3B = 123; 2% LC3B = 69; 4% LC3B = 66; 10% LC3B = 100; 50% LC3B = 93; 100% LC3B = 93.

The following figure supplements are available for Figure 3:

**Figure supplement 1.** (A) Quantification of the decay of the indicated mCherry-p62 variants from GST-LC3B-coated beads.

**Figure supplement 2.** p62 association to GST-LC3B-coated beads.

**Figure supplement 3.** Fluorescence recovery after photo-bleaching (FRAP) curves of mCherry-p62 delta PB1 recruited to beads coated with the indicated GST-LC3B concentrations.

**Figure supplement 4.** Steady-state binding of the indicated mCherry-p62 variants to beads coated with indicated GST-LC3B amounts.

interaction with LC3B. Finally, a third possibility would be that the PB1 domain allosterically enhances binding to LC3B.

To discriminate between these possibilities, we first measured the steady-state binding of wild-type and delta PB1 mCherry-p62 to GST-LC3B-coated beads. To this end, we recruited wild-type and delta PB1 mCherry-p62 at different concentrations to glutathione beads coated with GST-LC3B and imaged them by spinning disk microscopy at equilibrium. The mCherry-p62 LIR mutant was used as a negative control (*Figure 2F,G*).

The fluorescence signal on the beads correlated well with the protein concentration for both wild-type and delta PB1 mCherry-p62 (*Figure 2G*). However, the titration curve of the wild-type protein showed a steeper slope compared to the delta PB1 protein and approached a plateau above a concentration of 5 µM (*Figure 2F* and *Figure 2—figure supplement 3*). We could, therefore, estimate a half-maximal binding constant of 1.5 µM for wild-type mCherry-p62. It was impossible to estimate the half maximal binding constant for the delta PB1 mutant since at higher protein concentrations the fluorescence of the unbound protein rendered an accurate quantification of the bead-associated signal impossible.

The different shapes of the titration curves (*Figure 2—figure supplement 3*) suggested that the presence of the PB1 domain does not merely confer piggybacking of p62 molecules, but actively increases the overall affinity of p62 toward LC3B. This could either be due to an oligomerization-dependent increase in avidity or an allosteric effect on the intrinsic affinity of the LIR motif for LC3B.

## Oligomerization of p62 renders binding to LC3B-coated surfaces irreversible

To discriminate between these possibilities, we first performed fluorescence recovery after photo-bleaching (FRAP) experiments to determine the exchange rate of mCherry-p62 on GST-LC3B coated beads (*Figure 3A,B*). In fact, if the PB1 domain increases the avidity of p62 towards surface-localized LC3B via oligomerization, this would result in a lower off-rate of the wild-type protein compared to the non-oligomerizing delta PB1 p62. This would in turn translate into a slower fluorescence recovery for the wild-type protein. Indeed, while p62 delta PB1 readily recovered 5 min after bleaching, wild-type p62 showed almost no recovery within the same time frame (*Figure 3A*).

We confirmed this result by following the dissociation of the two mCherry-p62 variants from GST-LC3B beads (*Figure 3—figure supplement 1*). To this end, GST-LC3B-coated beads were incubated with wild-type mCherry-p62 or the delta PB1 mutant, diluted into empty buffer and imaged over time. While delta PB1 p62 started to dissociate immediately after dilution, the wild-type protein

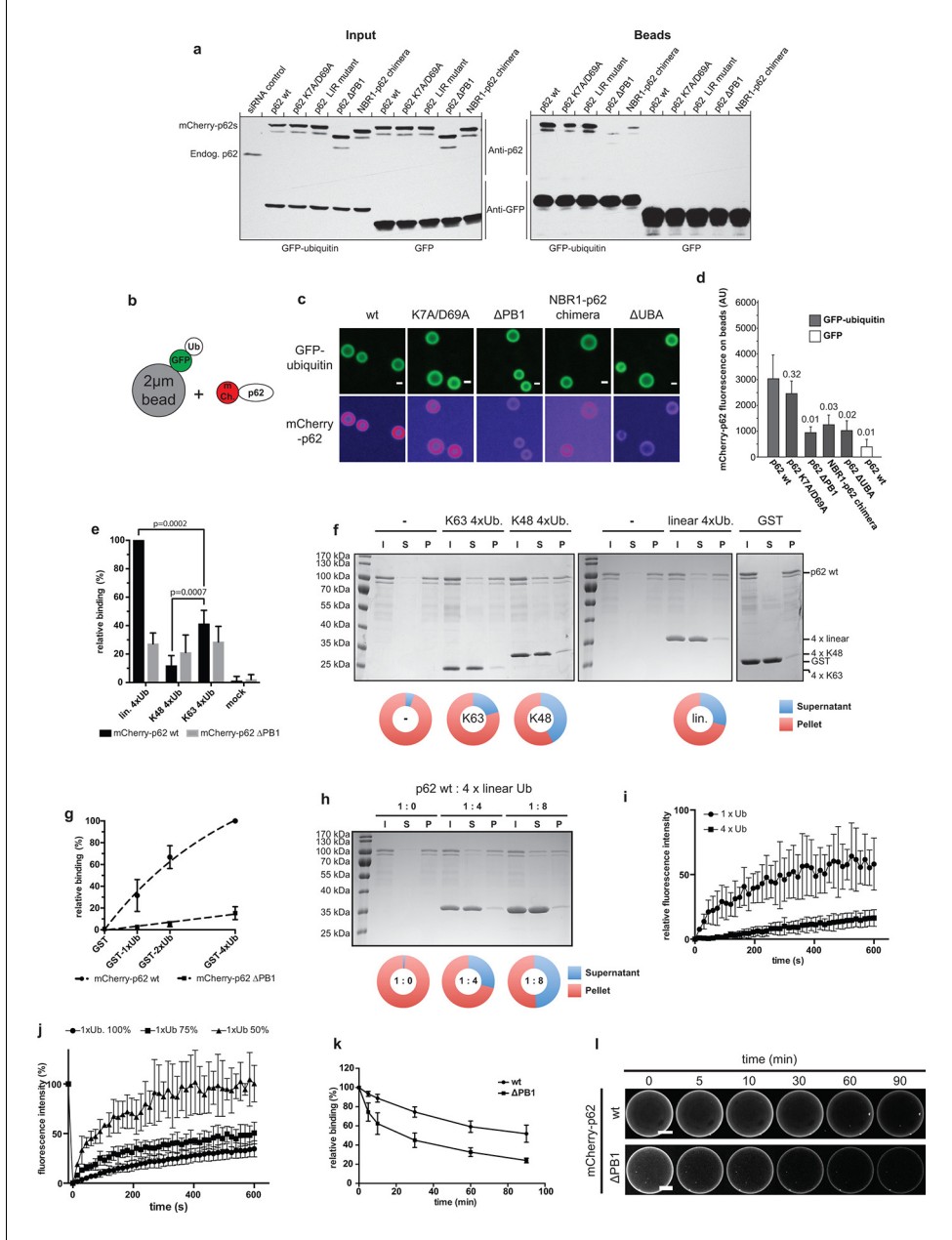

**Figure 4.** Oligomerization of p62 promotes ubiquitin binding. (A) GFP-TRAP experiment using HeLa cell lysates co-expressing GFP (control) or GFP-ubiquitin and the indicated mCherry-p62 variants. The endogenous p62 was silenced by siRNA treatment. Eight percent input and 100% of the bead fractions were analyzed by western blotting using anti-GFP and anti-p62 antibodies. (B) Scheme of the set-up of the experiment shown in (C) and (D) Recombinant GFP-ubiquitin was cross-linked to 2 µm latex beads and incubated with purified mCherry-p62 variants at 50 nM final concentration. Beads were observed using a spinning disk microscope under steady-state conditions. (C) Representative images of the recruitment of mCherry-p62 variants on GFP-ubiquitin-coated beads. Pictures were taken using the same microscopy settings and shown in false color for the mCherry-p62 signal (ImageJ: fire). Scale bar 1 µm. (D) Quantification of mCherry-p62 recruitment to beads coated with GFP-ubiquitin or GFP. Averages and SD of three independent replicates are shown. Indicated p-values were calculated with a two-tailed unpaired Student's *t*-test. (E) Quantification of steady-state binding of the indicated p62 variants to the indicated ubiquitin chains cross-linked to 2 µm latex beads. Averages and SD of three independent replicates are shown. All data are normalized to wild-type mCherry-p62 binding to linear tetra-ubiquitin. p-Values were calculated using a two-tailed unpaired Student's t-test.(F) Coomassie-stained gels showing p62 sedimentation assays conducted with recombinant wild-type mCherry-p62 in the presence of the indicated tetra-

*Figure 4. continued on next page*

*Figure 4. Continued*

ubiquitin chains. GST was used as a negative control. For each sample, the input, supernatant, and pellet fractions are shown. Quantifications are shown below the gel. The protein amount in the pellets and supernatants are expressed as fractions of the input. (**G**) Quantification of steady-state binding of the indicated p62 variants to beads coated with GST-mono-, di- or –tetra-ubiquitin. GST was used as negative control. Averages and SD of at least three independent experiments are shown. Data are normalized to wild-type mCherry-p62 binding to GST-tetra-ubiquitin. Data points were fitted to mono-exponential curves (dashed lines). (**H**) p62 co-sedimentation assay with increasing concentrations of linear tetra-ubiquitin. Wild-type mCherry-p62 was incubated with linear tetra-ubiquitin chains at the indicated molar ratios before ultracentrifugation. Inputs, supernatants and pellets were analyzed by SDS-PAGE followed by Coomassie staining. Quantification was performed as described for (**F**). (**I**) Fluorescence recovery after photo-bleaching (FRAP) curves of wild-type mCherry-p62 recruited to mono-ubiquitin or tetra-ubiquitin-coated beads. Averages and SD of six independent FRAP recordings are shown. (**J**) FRAP curves of wild-type mCherry-p62 recruited to beads coated with decreasing concentrations of mono-ubiquitin. For each sample, the averages and SD from six independent FRAP recordings are shown. (**K**) Quantification of wild-type and delta PB1 mCherry-p62 decay from GST-di-ubiquitin-coated beads. Averages and SD of three independent replicates are shown. (**L**) Representative images of data shown in (**K**). For better comparison, brightness was adjusted so that intensities of beads at time 0 is identical. Scale bars, 25 μm. (**D**) Total beads counted per condition: GFP-ub coated beads + mCherry-p62 wild-type = 565; GFP-ub coated beads + mCherry-p62 K7A/ D69A = 383; GFP-ub coated beads + mCherry-p62 delta PB1 = 378; GFP-ub coated beads + mCherry-NBR1-p62 chimera = 476; GFP-ub coated beads + mCherry-p62 LIR mutant = 393; GFP-ub coated beads + mCherry-p62 ΔUBA = 347; GFP coated beads + mCherry-p62 wild-type = 187.(**E**) Total beads quantified per condition: mCherry-p62 WT: M1 4xUB = 427; K48 4xUB = 332; K63 4xUB = 305; mock = 95. mCherry-p62 delta PB1: M1 4xUB = 266; K48 4xUB = 239; K63 4xUB = 226; mock = 75.(**G**) Total beads quantified per condition: mCherry-p62 wild-type: GST = 107; GST-mono-ubiquitin = 182; GST-di-ubiquitin = 149; GST-tetra-ubiquitin = 236. mCherry-p62 delta PB1) GST = 113; GST-mono-ubiquitin = 165; GST-di-ubiquitin = 134; GST-tetra-ubiquitin = 241. (**K**) Total beads quantified: wild-type = 83, delta PB1 = 65.

The following figure supplements are available for Figure 4:

**Figure supplement 1.** p62-LC3B co-sedimentation assay.

---

remained stably bound to the beads for up to 1.5 hr after dilution. We, therefore, concluded that oligomerization decreases the off-rate of p62 from surface-localized LC3B.

We also followed the kinetics of association of wild-type and delta PB1 mCherry-p62 to GST-LC3B-coupled beads. Both proteins showed an initially fast association with the beads (*Figure 3— figure supplement 2*, insert). However, while no further increase in bead-associated signal was observed for the delta PB1 protein, the wild-type mCherry-p62 further accumulated on the beads over the time course of 1 hr (*Figure 3—figure supplement 2*).

Next, we asked whether oligomerization would also positively affect binding of p62 to free LC3B. We, therefore, immobilized p62 on RFP-TRAP beads and added free GFP-LC3B. The recruitment of wild-type and delta PB1 p62 to RFP-TRAP beads was equally efficient (*Figure 3C*, black bars). To our surprise, mCherry-p62 delta PB1 was twice as efficient as the wild-type protein in recruiting free GFP-LC3B (*Figure 3C*, gray bars, and *Figure 3D*). We then went on to measure the decay of the GFP-LC3B signal from the beads upon dilution (*Figure 3E*). Here, GFP-LC3B readily dissociated from beads coated with both the oligomeric and non-oligomeric p62 variants, with no significant difference.

Taken together, these results suggest that oligomerization of p62 specifically promotes interaction with surface-localized, clustered LC3B by drastically reducing the off-rate of p62 from LC3B-coated surfaces. In contrast, oligomerization does not affect the intrinsic affinity of LIR motif for LC3B since binding to free LC3B is not enhanced.

This hypothesis predicts that the stability of oligomeric p62 on LC3B-coated surfaces should directly correlate with the density of LC3B on the surface. We tested this hypothesis by recruiting wild-type mCherry-p62 to beads coated with decreasing densities of GST-LC3B and measured the fluorescence recovery rates after bleaching (*Figure 3F*).

Strikingly, decreasing the density of LC3B on the beads resulted in faster recovery rates for wild-type p62 (*Figure 3F*). In contrast, the recovery rate of p62 delta PB1 was not affected when the density of LC3B on the beads was reduced even by a factor of 10 (*Figure 3—figure supplement 3*).

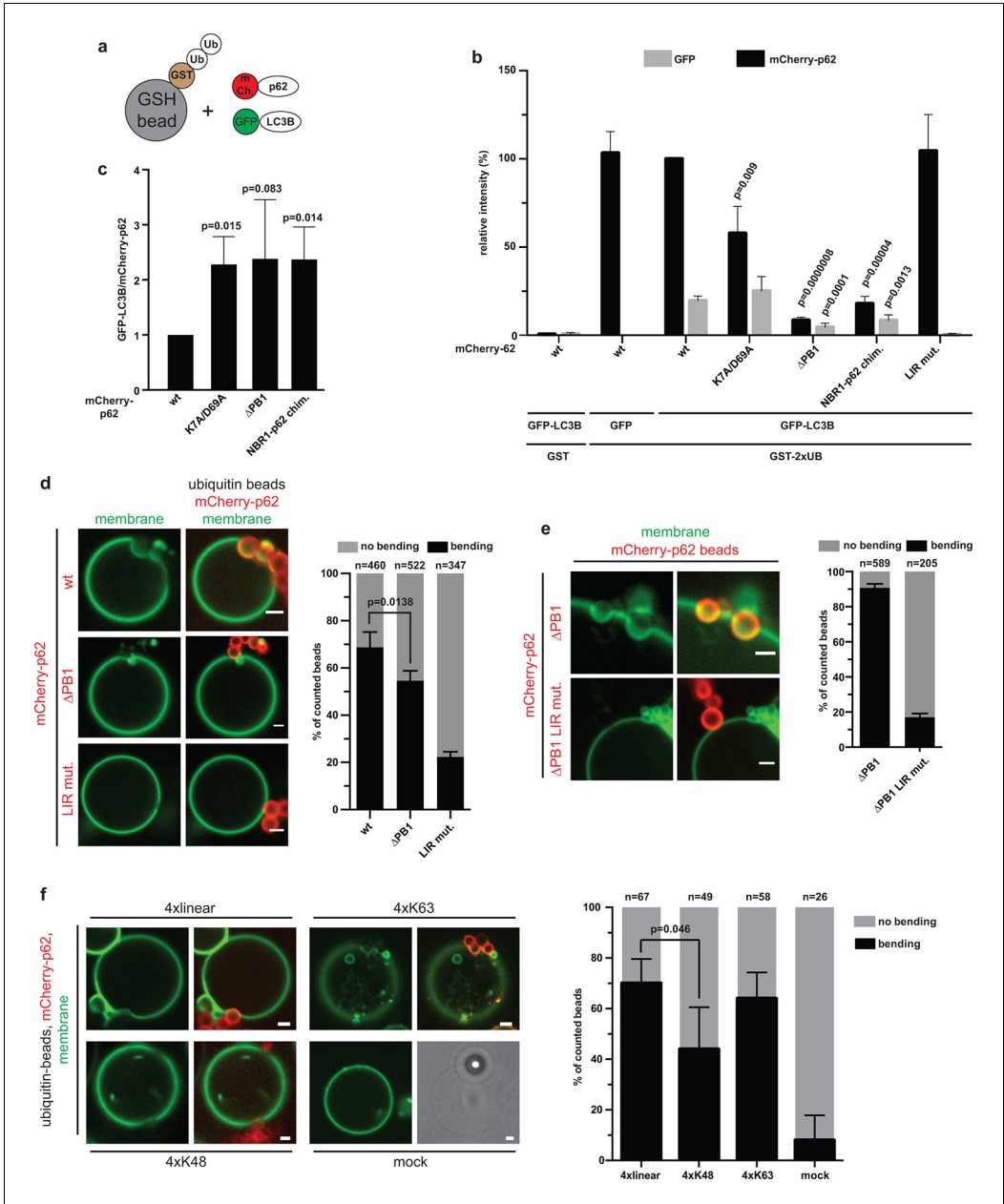

**Figure 5.** Reconstitution of p62–mediated membrane bending. (**A–C**) Indirect recruitment of GFP-LC3B to GST-di-ubiquitin coated beads via mCherry-p62. (**A**) Scheme of the experiment. GST-di-ubiquitin was pre-recruited to glutathione agarose beads. Beads were co-incubated with mCherry-p62 variants and GFP or GFP-LC3B. Imaging was performed at equilibrium. (**B**) Quantification of mCherry and GFP intensities on the beads (see *Figure 5—figure supplement 1* for representative pictures). All values are plotted as percentages of the wild-type mCherry-p62 intensity. Averages and SD of four independent replicates are shown. Indicated p-values were calculated with a two-tailed unpaired Student's t-test. p-Values above black bars refer to the mCherry-p62 wild-type bar; p-values above gray bars refer to the GFP-LC3B intensity in the wild-type mCherry-p62 sample. (**C**) Plot of GFP/mCherry ratio of data shown in (**B**). The ratio for wild-type mCherry-p62 was normalized to 1. All p-values were calculated with a two-tailed unpaired Student's t-test. (**D**) Quantification and representative pictures of LC3B-positive giant unilamellar vesicle (GUV) membranes bending around 2 µm glutathione beads coated with GST-tetra-ubiquitin and incubated with the indicated mCherry-p62 variants. Averages and SD of four independent experiments are shown. The indicated p-value was calculated with a two-tailed unpaired Student's t-test. *n* numbers indicate the total number of beads quantified per sample. Scale bars, 2 µm. (**E**) Quantification and representative pictures of LC3B-positive GUV membranes bending around 2 µm latex beads cross-linked with the indicated mCherry-p62 variants. Averages and SD of three independent experiments are shown. *n* numbers indicate the total number of beads quantified per sample. Scale bars, 2 µm. (**F**) Quantification and representative pictures of LC3B-positive GUV membranes bending around 2 µm latex beads cross-linked with the indicated ubiquitin chains and incubated with wild-type mCherry-p62. Averages and SD of four independent experiments are shown. The indicated p-value was calculated with a two-tailed unpaired Student's t-test. *n* numbers indicate the total

*Figure 5. continued on next page*

*Figure 5. Continued*

amount of beads counted per sample. Scale bars, 2 µm. (B) Total beads counted per condition: GST + mCherry-p62 wild type + GFP-LC3B = 101, GST-2xUB + mCherry-p62 wild type + GFP = 125, GST-2xUB + mCherry-p62 wild type + GFP-LC3B = 174, GST-2xUB + mCherry-p62 DM + GFP-LC3B = 172, GST-2xUB + mCherry-p62 delta PB1 + GFP-LC3B = 154, GST-2xUB + mCherry-NBR1-p62 chimera + GFP-LC3B = 153, GST-2xUB + mCherry-p62 LIR mut + GFP-LC3B = 129.

The following figure supplements are available for Figure 5:

**Figure supplement 1.** Representative pictures of the data shown in *Figure 5B*.

We then plotted the recovery rates extrapolated from the FRAP curves against the respective LC3B density on the beads (*Figure 3G*). The data points fitted robustly to an exponential curve, which showed a half-maximum around 14% of LC3B density. This value is in line with the result we obtained when we measured the steady-state binding of p62 to beads coated with different densities of GST-LC3B ($c_{1/2}$ = 9.5% for the wild-type protein, *Figure 3—figure supplement 4*).

The data above strongly support a model of oligomerization-dependent LIR motif clustering and hence high-avidity interactions with surfaces on which LC3B is clustered. If this is indeed the case, then the same behavior should be displayed by a non-oligomerizing version of p62 containing multiple LIR motifs.

We, therefore, generated a mCherry-p62 delta PB1 protein containing 4 LIR motifs (4xLIR) and tested its exchange rate on LC3B-coated beads by FRAP (*Figure 3H,I*). Strikingly, p62 delta PB1 4xLIR showed a recovery rate approximately four times slower than delta PB1 p62 containing only one LIR motif (*Figure 3H*).

## Oligomerization of p62 promotes the interaction with ubiquitin and confers chain specificity

Given the effect of p62 oligomerization on LC3B binding, we asked whether a similar mechanism applied to the interaction with ubiquitin. Indeed, it was previously reported that the deletion of the PB1 domain resulted in reduced interaction with ubiquitin in a pull-down assay (*Kirkin et al., 2009*). We first tested the interaction of mCherry-p62 with GFP-ubiquitin in GFP-TRAP experiments using cell lysates from transfected HeLa cells (*Figure 4A*) in which the endogenous p62 was downregulated by siRNA treatment. The ability of the p62 variants to co-precipitate with GFP-ubiquitin correlated strongly with their ability to oligomerize. While the wild-type protein and the LIR mutant interacted most robustly with ubiquitin, this interaction was markedly reduced for the K7A/D69A mutant. The non-oligomerizing delta PB1 mutant and the NBR1-p62 chimera showed barely detectable interactions with ubiquitin.

Next, we investigated whether the same was true under equilibrium conditions. To this end, we covalently coupled GFP-ubiquitin to 2 µm beads and added wild-type mCherry-p62 at a final concentration of 50 nM (*Figure 4B–D*). Spinning disk microscopy was then used to determine the association of p62 with the beads. The non-oligomeric delta PB1 mutant and the NBR1-p62 chimera as well as the ΔUBA mutant showed strongly reduced recruitment to the GFP-ubiquitin-coated beads when compared to the wild-type protein (*Figure 4C,D*). Consistent with the GFP-TRAP experiment (*Figure 4A*), the K7A/D69A mutant showed only slightly reduced binding to the GFP-ubiquitin-coated beads.

In vivo individual ubiquitin molecules are frequently covalently attached to one another forming longer chains. Depending on the residue used for the linkage formation, different chain types can be formed, each of them with a different functional role in the cell (*Husnjak and Dikic, 2012*; *Komander and Rape, 2012*). p62 was shown to bind preferentially K63- over K48-linked chains (*Long et al., 2008*; *Matsumoto et al., 2011*; *Seibenhener et al., 2004*). We, therefore, asked whether oligomerization of p62 influences the binding specificity for different ubiquitin chains.

To this end, we cross-linked linear (M1)-, K48-, or K63-linked tetra-ubiquitin chains to 2 µm beads and measured the binding of wild-type and delta PB1 mCherry-p62 at equilibrium (*Figure 4E*, black bars). Consistent with previous reports, wild-type mCherry-p62 bound stronger to K63-linked chains than to the K48-linked chains (*Long et al., 2010*; *Matsumoto et al., 2011*; *Seibenhener et al., 2004*). The strongest binding was detected for linear ubiquitin. When we compared the binding

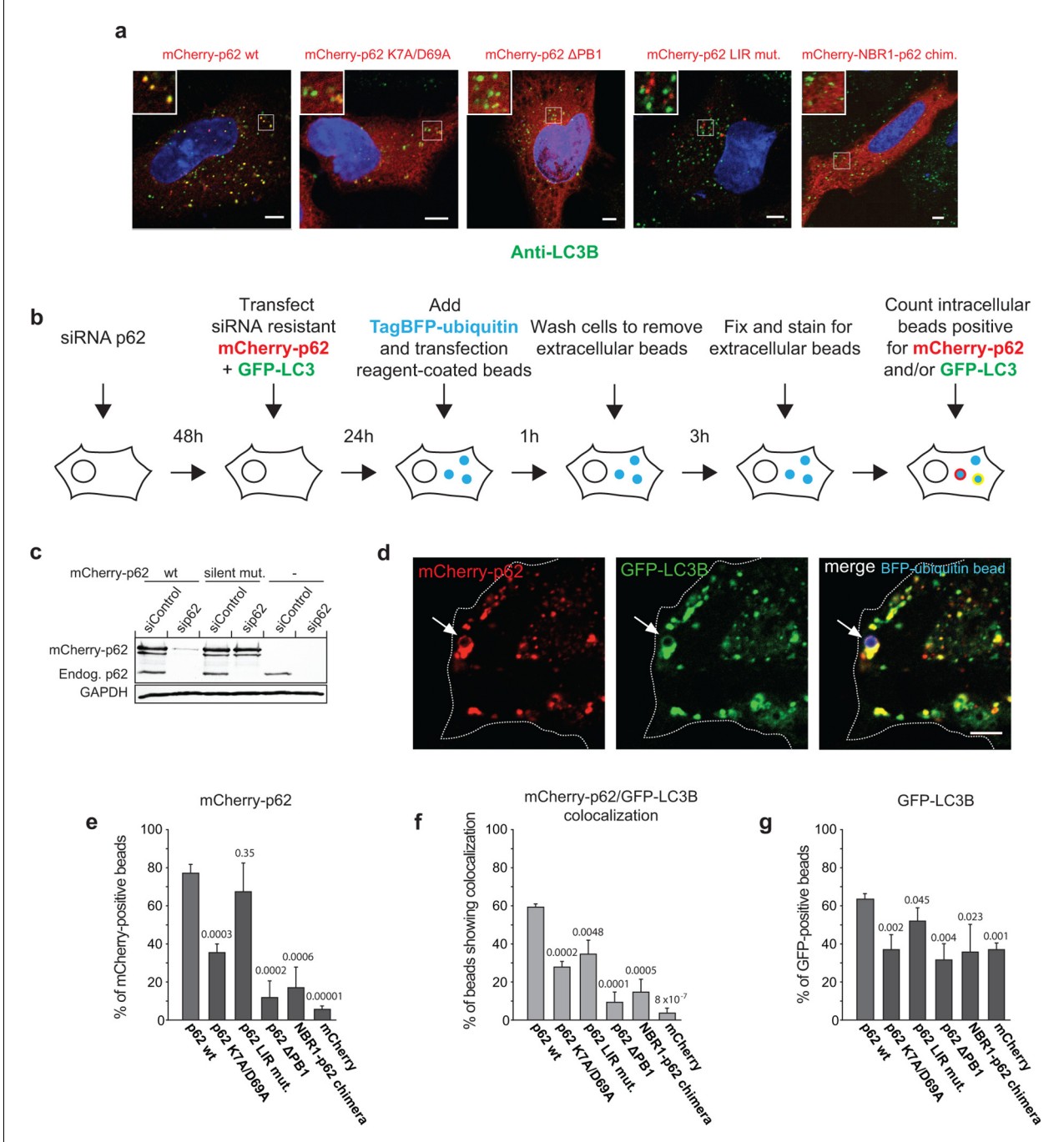

**Figure 6.** Oligomerization of p62 promotes recruitment of p62 and LC3B to ubiquitin-coated beads in HeLa cells. (**A**) Anti-LC3B immunofluorescence analysis of HeLa cells transiently transfected with mCherry-p62 variants. Nuclei were stained with DAPI. Insets show magnifications of the indicated squares. Scale bars, 5 μm. (**B–G**) Quantification of mCherry-p62 and GFP-LC3B recruitment around artificial cargo particles in HeLa cells. (**B**)Schematic outline of the experiment. (**C**) Western blot analysis of HeLa cell lysates overexpressing wild–type mCherry-p62 with or without silent mutations in the siRNA targeting region. (**D**) HeLa cell co-expressing siRNA resistant wild-type mCherry-p62 and GFP-LC3B. Endogenous p62 was silenced by siRNA (see **Figure 6—figure supplement 1**). The arrows indicate co-localization of mCherry-p62 and GFP-LC3B at a BFP-ubiquitin-coated 2 μm bead. Scale bar: 5 μm. (**E**) Quantification of mCherry-p62 variants localizing to BFP-ubiquitin-coated beads in mCherry-p62 and GFP-LC3B co-expressing cells. (**F**) Quantification of co-localization of mCherry-p62 variants and GFP-LC3B at BFP-ubiquitin-coated beads. (**G**) Quantification of GFP-LC3B localization to BFP-ubiquitin-coated beads. For all data in (**D–G**), averages and SD of three independent replicates are shown. Indicated p-values were calculated by a two-tailed equal-variance Student's *t*-test. All graphs show the averages and SD. (**E–G**) Total beads quantified per condition: wild-type = 113 beads, K7A/D69A = 145 beads, delta PB1 = 117 beads, NBR1-p62 chimera = 120 beads, mCherry = 144 beads.

*Figure 6. continued on next page*

*Figure 6. Continued*

The following figure supplements are available for Figure 6:

**Figure supplement 1.** (A) Representative picture of 2 µm latex beads cross-linked with BFP-ubiquitin.

intensities of p62 delta PB1 (*Figure 4E*, gray bars) with the wild-type protein, we made two observations: first, binding to linear ubiquitin was strongly reduced, and second, there was no longer a significant difference in binding to the three chain types. We concluded that oligomerization of p62 determines specificity toward linear and perhaps weakly toward K63-linked ubiquitin chains, while non-oligomerizing p62 delta PB1 binds indifferently to all three chain types. Interestingly, oligomerization does not promote binding to K48-linked ubiquitin chains.

It was reported that addition of K63-linked ubiquitin chains partially disrupted p62 oligomers (*Ciuffa et al., 2015*). Employing the p62 pelleting assay ([*Ciuffa et al., 2015*] and *Figure 2B,C*), we tested the effect of linear, K48- and K63-linked tetra-ubiquitin chains on the oligomerization of p62. All three chain types had a measurable effect on the oligomerization of p62 (*Figure 4F*). This effect was specific as GST did not disrupt p62 oligomers. Consistent with previous experiments (*Ciuffa et al., 2015*), we did also not detect any effect of LC3B on the oligomerization of p62 [*Figure 4—figure supplement 1*]. Addition of K48-linked ubiquitin chains had the strongest disruptive effect on p62 oligomerization (*Figure 4F*). Together with the fact that these chains were not preferentially bound by oligomeric p62 (*Figure 4E*) this suggests that p62 oligomers may be locally disrupted upon binding to the beads cross-linked with K48-linked ubiquitin chains.

Since the strongest oligomerization-dependent binding of p62 to ubiquitin was detected for linear chains, we analyzed this interaction further. When beads coupled to mono-ubiquitin, linear di-ubiquitin, and tetra-ubiquitin were tested for p62 binding, it became apparent that both wild-type and delta PB1 p62 bound stronger to longer ubiquitin chains (*Figure 4G*). Thus, even though linear ubiquitin chains disrupt p62 oligomers to some extent in a concentration-dependent manner (*Figure 4H*), they are still bound stronger than mono-ubiquitin (*Figure 4G*). We then went on to study whether lower off-rates contribute to the stronger binding of the wild-type protein to linear tetra-ubiquitin compared to mono-ubiquitin. Indeed, FRAP analysis of wild-type mCherry-p62 bound to mono-ubiquitin and linear tetra-ubiquitin showed that the recovery rate was higher for mono-ubiquitin (*Figure 4I*). Lowering the density of mono-ubiquitin on the beads also resulted in increased FRAP recovery rates (*Figure 4J*), similarly to what we observed for LC3B (*Figure 3F*). These results suggested that oligomerization of p62 results in clustering of the ubiquitin-binding UBA domain and thus avid binding to surface-localized, clustered ubiquitin, analogous to the interaction with surface-localized LC3B (*Figures 2 and 3*). To directly test whether oligomerization of p62 confers more avid interaction with ubiquitin on surfaces, we measured the decay of mCherry-p62 wild-type or delta PB1 from ubiquitin-coated beads upon dilution in buffer (*Figure 4K,L*). Both proteins showed some degree of dissociation from the surface of the beads but the oligomeric wild-type p62 remained more stably bound.

## p62 drives membrane bending around cargo particles

In vivo p62 interacts with ubiquitin when it is concentrated on the cargo and with LC3B when it is localized on the isolation membrane. We, therefore, asked what the consequences of the simultaneous interaction of p62 with ubiquitin and LC3B would be in the context of membrane-localized LC3B and ubiquitin localized to a surface. First, we asked whether p62 actually possesses the ability to simultaneously interact with LC3B and ubiquitin. To this end, we conducted experiments using GST-di-ubiquitin-coated beads to indirectly recruit GFP-LC3B via p62 (*Figure 5A–C*, *Figure 5—figure supplement 1*). First, the recruitment of mCherry-p62 variants to GST-di-ubiquitin-coated beads recapitulated our results with GFP-mono-ubiquitin (compare *Figure 5B*, black bars, with *Figure 4D*). We next assessed the ability of the p62 variants to recruit GFP-LC3B to the beads. No significant difference between wild-type p62 and the K7A/D69A mutant was observed, while p62 delta PB1 and the NBR1-p62 chimera recruited significantly less LC3B (*Figure 5B*, gray bars). However, when the GFP-LC3B signal is normalized to the corresponding mCherry-p62 signal, the non-oligomerizing

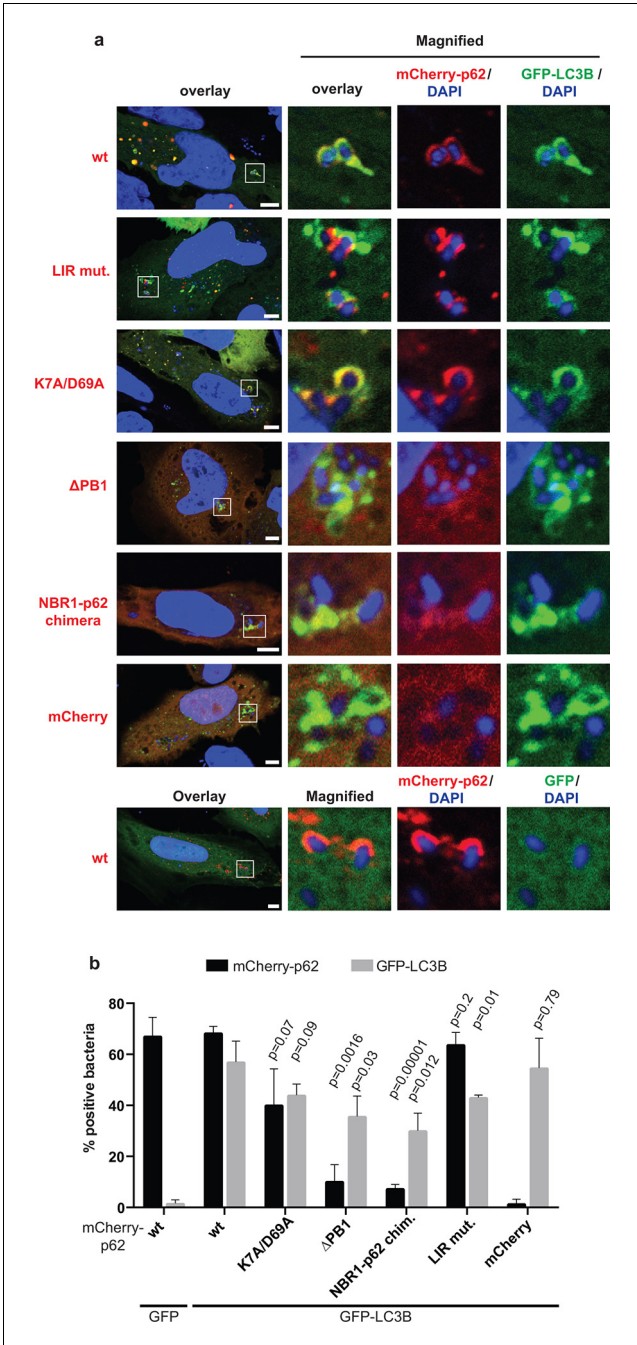

**Figure 7.** Oligomerization of p62 is required for efficient recruitment of p62 to *Salmonella typhimurium* in HeLa cells. (A) Representative pictures of HeLa cells co-expressing GFP-LC3B and mCherry-p62 infected with *S. typhimurium*. The endogenous p62 was silenced by siRNA. Magnifications of the insets are shown on the right. Pictures of whole cells are shown in *Figure 7—figure supplement 1*. Scale bars, 5 μm. (B) Quantification of mCherry-p62- and/or GFP-LC3B-positive bacteria. Averages and SD of three independent replicates are shown. Indicated p-values were calculated with a two-tailed unpaired Student's *t*-test. Values above the black bars refer to the wild-type mCherry-p62 value in the wild-type mCherry-p62 + GFP-LC3B sample; values above the gray bars refer to the GFP-LC3B value in the same sample. (B) Total bacteria counted per condition: mCherry-p62 wild-type + GFP-LC3B = 245, mCherry-p62 K7A/D69A + GFP-LC3B = 337, mCherry-p62 LIR + GFP-LC3B = 287, mCherry-p62 delta PB1 + GFP-LC3B = 296, mCherry-NBR1-p62 chimera + GFP-LC3B = 292, mCherry + GFP-LC3B = 318, mCherry-p62 wild-type + GFP = 325

*Figure 7. continued on next page*

*Figure 7. Continued*

The following figure supplements are available for Figure 7:

**Figure supplement 1.** Full-size pictures of the data shown in *Figure 7A*.

**Figure supplement 2.** Quantification of co-localization of the indicated mCherry-p62 variants and GFP-LC3B at bacteria.

mutants appear to be about twice as efficient as p62 wild-type in recruiting GFP-LC3B (*Figure 5C*). This mirrors the results we obtained when we directly tethered mCherry-p62 to RFP-TRAP beads (*Figure 3D*).

In summary, oligomerization of p62 generates high-avidity interactions that increase the residence time of the oligomeric particle on LC3B and ubiquitin-coated structures. However, the efficiency of interaction with LC3B for each p62 monomer within the oligomeric structure is reduced.

In order to more fully reconstitute the system in vitro, we attached LC3B-6xHis to the surface of GUVs (*Figure 5D*). To visualize the vesicles, the membrane was labeled by incorporation of Oregon-green phosphatidylethanolamine. GST-linear tetra-ubiquitin was bound to 2 μm glutathione beads. The beads were then incubated with mCherry-p62 variants and added to GUVs. Strikingly, wild-type mCherry-p62 mediated strong bending of the GUV membrane around the beads. Frequently, the beads were completely submerged into the GUVs (*Figure 5D*). Membrane bending was dependent on the specific interaction with LC3B as the LIR mutant showed strongly reduced membrane bending activity. The non-oligomerizing delta PB1 mutant showed reduced membrane-bending efficiency, likely due to the fact that less p62 delta PB1 was localized to the ubiquitin-coated beads (*Figure 4*).

To determine whether membrane bending by p62 also specifically required its interaction with ubiquitin, we directly cross-linked mCherry p62 delta PB1 to beads. Cross-linked p62 delta PB1 efficiently mediated membrane bending in a LIR-dependent manner (*Figure 5E*), showing that the presence of ubiquitin is not essentially required for membrane bending. Next, we tested the ability of p62 to bend the membrane around beads cross-linked to linear, K48- and K63-linked tetra-ubiquitin chains (*Figure 5F*). We observed membrane bending events for all the three chain types. However, membrane bending was significantly reduced for K48-linked ubiquitin, consistent with the lower affinity of p62 for this chain (*Figure 4E*).

In summary, we conclude that the interaction of p62 with ubiquitin and LC3B is sufficient to drive bending of a LC3-coated membrane around an ubiquitin-positive cargo.

## Oligomerization of p62 promotes its relocalization to cargo and LC3B recruitment

Given our in vitro results, we wanted to know whether oligomerization mutants of p62 would localize to LC3B-positive structures in vivo. We, therefore, examined the co-localization of mCherry-p62 with endogenous LC3B in HeLa cells in which the endogenous p62 was silenced by siRNA (*Figure 6A*). Consistent with earlier results (*Bjørkøy et al., 2005*; *Ichimura et al., 2008*; *Pankiv et al., 2007*) the wild-type and LIR mutant proteins localized in multiple *puncta*, but only wild-type p62 extensively co-localized with LC3B. The delta PB1 protein and the NBR1-p62 chimera showed no *puncta* formation and appeared cytosolic. Interestingly, the K7A/D69A protein was largely cytosolic but still displayed some degree of *puncta* formation and co-localization with LC3B (*Figure 6A*).

Next, we went on to test whether the ability of p62 to oligomerize would affect its accumulation around cargo particles and its ability to recruit LC3B also in cells. To this end, we adapted a previously described assay that is based on the coating of small latex beads with transfection reagent ([*Kobayashi et al., 2010*] and *Figure 6B*). Upon internalization of the beads by the cell, the transfection reagent damages the endosomal membrane, which then becomes a target for selective autophagy (*Kobayashi et al., 2010*; *Thurston et al., 2012*). In order to render the beads themselves a direct target for selective autophagy, we coated them with recombinant TagBFP-ubiquitin before coating with transfection reagent (*Figure 6—figure supplement 1A*). TagBFP-ubiquitin-coated beads were then added to HeLa cells that had the endogenous p62 protein downregulated by RNAi (*Figure 6C* and *Figure 6—figure supplement 1B*) and that were co-transfected with mCherry-p62

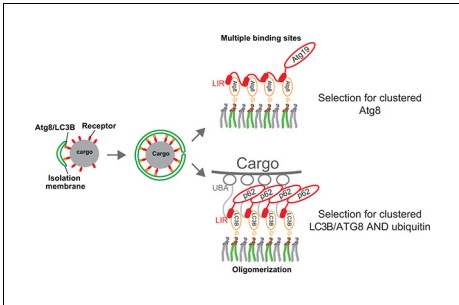

**Figure 8.** A model for selective autophagy in yeast and mammalian cells. Multiple binding sites in the yeast cargo receptor Atg19 promote selective and exclusive engulfment of cargo material by Atg8-covered membranes (*Sawa-Makarska et al., 2014*). Oligomerization of p62 allows it to simultaneously select for clustered ubiquitin and ATG8-family proteins.

and GFP-LC3B. Extracellular beads were stained using an anti-ubiquitin antibody allowing us to count only the intracellular beads (*Figure 6—figure supplement 1C*). mCherry-p62 wild–type was robustly recruited to the TagBFP-ubiquitin-coated beads (*Figure 6D,E*). The ability of the p62 mutants to associate with the beads strongly correlated with their ability to oligomerize (*Figure 6E*). While the non-oligomeric delta PB1 mutant and the NBR1-p62 chimera showed almost no recruitment above the experimental background (mCherry), the K7A/D69A mutant was still recruited to a considerable degree (*Figure 6E*). To follow the recruitment of LC3B to the beads, we quantified the number of beads that were positive for both mCherry-p62 and GFP-LC3B (*Figure 6D,F*). The recruitment of GFP-LC3B to the beads was also strongly dependent on the ability of p62 to oligomerize. Wild-type p62 showed robust recruitment of LC3B to the beads while p62 delta PB1 and the NBR1-p62 chimera showed very low LC3B recruitment. The p62 K7A/D69A mutant displayed an intermediate behavior between the two extremes with regard to LC3B recruitment. Moreover, we noticed that the effect of the oligomerization on the LC3B recruitment to the beads was largely dependent on the reduced recruitment of the p62 oligomerization mutant. This became obvious when we quantified the total recruitment of LC3B to all intracellular beads, regardless of whether they were positive or negative for p62 (*Figure 6G*). This quantification showed that there is at least one redundant factor that is able to recruit LC3B to the ubiquitin-coated beads. This redundant factor accounts for 68% of the LC3B recruitment activity (compare wild-type p62 to mCherry). Obvious candidates for this factor are other cargo receptors such as NBR1 (*Kirkin et al., 2009*), optineurin (*Wild et al., 2011*), or Tollip (*Lu et al., 2014*). However, within the dynamic range of our assay for p62 recruitment activity, the p62 oligomerization mutants showed a profound loss of LC3B recruitment.

In conclusion, the experiments presented in *Figure 6B–G* show that both the recruitment of p62 to ubiquitin-positive beads and the recruitment of LC3B to these beads by p62 are promoted by oligomerization of p62.

We next extended our analysis to a more physiological target of p62 and infected HeLa cells with *Salmonella typhimurium*, an intracellular pathogenic bacterium previously shown to be a p62 target (*Zheng et al., 2009*) (*Figure 7* and *Figure 7—figure supplement 1*).

LC3B was efficiently recruited to the bacteria even in the absence of p62 (*Figure 7B*, gray bars). This suggests that there might be other mechanisms for the autophagic targeting of intracellular *Salmonella*.

Wild-type p62 was robustly recruited to intracellular *Salmonella* and showed extensive co-localization with LC3B in vicinity of the bacteria. The LIR mutant also showed robust recruitment to the *Salmonella*. However, in contrast to the HeLa cells expressing wild-type p62, the LIR mutant and LC3B were localized into different patches, showing only partial co-localization (*Figure 7A*, *Figure 7—figure supplement 1,2*). The K7A/D69A mutant was still recruited to the bacteria while the oligomerization-deficient delta PB1 mutant and the NBR1-p62 chimera were not robustly recruited to the *Salmonella* (*Figure 7B*). Furthermore, all of these mutants showed reduced co-localization with LC3B (*Figure 7B*, *Figure 7—figure supplement 2*), similar to what we observed for ubiquitin-positive latex beads (*Figure 6F,G*).

Interestingly, when we quantified the overall amount of bacteria positive for LC3B, regardless of the presence of p62, we could see a mild dominant negative effect of p62 oligomerization mutants (*Figure 7B*, gray bars).

## Discussion

In this study, we have shown that the human cargo receptor p62 employs oligomerization to generate high-avidity interactions with ubiquitin and LC3B. Thus, oligomerization enables p62 to simultaneously select for concentrated ubiquitin and LC3B (*Figure 8*). There are interesting parallels but also deviations compared to the yeast Atg19 cargo receptor. Atg19 binds its prApe1 cargo with very high affinity and selects for membrane-bound Atg8 via a high-avidity interaction mediated by multiple low-affinity Atg8 interaction sites (*Sawa-Makarska et al., 2014*). These properties are advantageous because the prApe1 cargo is a dedicated selective autophagic cargo that needs to be delivered into the vacuole in order to fulfill its function (*Klionsky et al., 1992*). Thus, the high-affinity interaction of Atg19 with the prApe1 cargo ensures its rapid transport into the vacuole. The multiple Atg8 binding sites in Atg19 subsequently mediate the selective interaction with membrane localized, locally concentrated Atg8, enabling Atg19 to bend the membrane tightly around the cargo and to exclude non-cargo material from its delivery into the vacuole (*Baba et al., 1997*; *Sawa-Makarska et al., 2014*). In contrast, the cargo of p62 is not normally destined to be transported into the lysosomal system but fulfills a function in the cell's cytoplasm. Only when this material becomes dysfunctional or superfluous it becomes marked with ubiquitin and thereby a target for selective autophagy. Aggregated proteins, for example, are a cargo for p62 (*Bjørkøy et al., 2005*). However, when cytosolic proteins unfold and aggregate, they initially become a target for the ubiquitin-proteasome system (UPS). When the UPS is overwhelmed, unfolded, ubiquitinated proteins form aggregates on which ubiquitin is locally concentrated. Only these structures should become a target for p62 and subsequently be degraded by selective autophagy. The low affinity but high avidity interaction of the p62 oligomer with ubiquitin will select for these structures as ubiquitin is locally concentrated on them (*Figure 8*). Interestingly, we found that K48-linked ubiquitin chains are less efficiently bound by oligomeric p62 compared to linear and K63-linked chains, possibly because K48-linked ubiquitin chains disrupt the p62 oligomers more effectively. These results hint to the possibility that K48-linked chains are not the preferred target for p62 in vivo, possibly in order to prevent proteins targeted for the proteasome to become premature targets of p62. Thus aggregated proteins may need further modification by K63-linked or linear ubiquitin chains in order to render them efficient targets for p62.

Our data show that oligomerization has no direct effect on the individual LC3B–LIR and ubiquitin–UBA interactions. Instead, oligomerization drastically increases the residence time of p62 on LC3B and ubiquitin-coated surfaces. In the case of wild-type p62 and concentrated LC3B, the interaction even becomes irreversible and might represent the end point of a pathway in which the whole structure is eventually degraded in the lysosome.

Moreover, we report that non-oligomerizing p62 mutants are more efficient than wild-type p62 in recruiting soluble LC3B. Consistent with a recently published structure of p62, which showed a helical arrangement of the oligomer (*Ciuffa et al., 2015*), we hypothesize that p62 oligomers adopt a three-dimensional structure that does not allow all LIR motifs to be engaged in LC3B interactions at the same time. Furthermore, we speculate that in vivo oligomerization is required to tightly appose the growing LC3B-positive membranes to the cargo particle, largely due to the fact that oligomerization mediates the concentration of p62 at the cargo.

Additional regulation by cargo-localized kinases such as TBK1 (*Matsumoto et al., 2011*; *2015*) may generate positive feedback loops resulting in more efficient delivery of ubiquitinated cargo into the lysosome. Furthermore, cooperation of p62 with other cargo receptors such as NBR1 fine-tunes the process and contributes to the clustering of aggregated proteins into larger structures (*Kirkin et al., 2009*).

Interestingly, we found that oligomerization mutants of p62 showed a dominant negative effect on LC3B recruitment around *Salmonella*. It is possible that the presence of p62 oligomerization mutants prevents other cargo receptors or the autophagic machinery in general to trigger the formation of an LC3B-positive isolation membrane.

Self-association of autophagic cargo receptors in order to generate high-avidity interaction surfaces to select ubiquitinated cargo and ATG8-family protein decorated membranes may be a reoccurring theme. The cargo receptor optineurin forms higher-order oligomers (*Gao et al., 2014*), while the NBR1 cargo receptor dimerizes (*Kirkin et al., 2009*). It would, therefore, be interesting to test whether a similar molecular mechanism is also employed by these receptors.

## Materials and methods

### Accession numbers

p62/SQSTM1: NP_003891; MAP1LC3B (LC3B): NP_073729; GABARAP: NP_009209.1; ubiquitin: NP_001268649; NBR1: AAH09808

### Protein expression and purification

6xHis-TEV-mCherry-p62 constructs were generated as follows: human p62 was first cloned into pmCherry-C1 (Clontech, Mountain View, CA, USA) and then the mCherry-p62 fusion was subcloned into pET-Duet1. A Tobacco Etch Virus (TEV) protease recognition site was also included to remove the 6xHis-tag. The K7A/D69A, LIR (DDDW335-338AAAA), delta PB1 ($\Delta$2-102), delta PB1 LIR mutant and $\Delta$UBA ($\Delta$389-434) mutants were generated by PCR-based mutagenesis.

The NBR1-p62 chimera was generated as follows: a fragment coding for amino acids 1–85 from human NBR1 was cloned into pET-Duet1, followed by insertion of a fragment coding for amino acids 103–443 of p62. The NBR1-p62 chimera was subcloned into pmCherry-C1 and the whole mCherry-NBR1-p62 construct was finally cloned into pET-Duet1 to generate 6xHis-TEV-mCherry-NBR1-p62. The TEV site was used to remove the 6xHis-tag.

p62 delta PB1 4xLIR was generated as follows: pETDuet-6xHis-TEV-mCherry-p62 delta PB1 was used as template for a PCR reaction using a forward primer with an overhang coding for the GSGSS-GGDDDWTHLSS amino acid sequence. Upon self-ligation, the resulting construct coded for a 2xLIR version of p62 with the amino acids 332–343 (SGGDDDWTHLSS [numbers relative to the wild-type protein]) inserted after the wild-type LIR motif (after amino acid 343). The primer inserted an additional GSGS spacer between the two LIRs. p62 delta PB1 2xLIR was used as template for another PCR reaction that introduced *Hin*dIII and *Sal*I sites between LIR1 and LIR2. After self-ligation of this PCR product, an oligo coding for two LIR motifs (GSSGGDDDWTHLSS) was inserted via the *Sal*I and *Hin*dIII sites. The final 6xHis-TEV-mCherry-p62 delta PB1 4xLIR construct coded for a protein with the following sequence inserted between amino acids 343 and 344: GSGSSGG<u>DDDWTHL</u>SSGSSGG<u>DDDWTHL</u>SSGSSGG<u>DDDWTHL</u>SS (the numbers are relative to the wild-type protein and the additional three core LIR motifs are underlined).

The proteins were expressed in *Escherichia coli* Rosetta (DE3) pLysS cells. Bacteria were grown in Luria broth (LB) medium until OD$_{600}$ ≈ 0.8–1, induced with 0.1 mM isopropylthiogalactoside (IPTG) and grown at 25°C for 5 hr. Harvested cells were resuspended in lysis buffer 50 mM 4-(2-hydroxyethyl)-1-piperazineethanesulfonic acid (HEPES) at pH 7.5, 500 mM NaCl, 10 mM imidazole, 2 mM MgCl$_2$, 2 mM β-mercaptoethanol, complete protease inhibitor (Roche, Basel, Switzerland) and DNase I and lysed by a freeze–thaw cycle followed by brief 30 s sonication.

Lysates were cleared by ultracentrifugation at 140,000 *g* for 30 min at 4°C (Beckman, Brea, CA, USA, Ti45 rotor). Supernatants were applied to Ni-NTA columns (GE Healthcare, Buckinghamshire, UK) and 6xHis-tagged p62 constructs were eluted via a stepwise imidazole gradient (50, 75, 100, 150, 200, and 300 mM). Protein-containing fractions were pooled and subjected to overnight cleavage with TEV protease at 4°C.

Cleaved proteins were applied to a Superdex 200 column (16/600, GE Healthcare) and eluted with a buffer containing 25 mM HEPES pH 7.5, 500 mM NaCl and 1 mM dithiothreitol (DTT). Fractions containing the purified proteins were pooled, concentrated, frozen in liquid nitrogen, and stored at -80°C.

GST-LC3B was generated by insertion of the human LC3B coding sequence into pGEX-4T1. The last five amino acids of LC3B were deleted to mimic Atg4 cleavage. GST-di-ubiquitin and tetra-ubiquitin plasmids were a courtesy of Fumiyo Ikeda, Vienna, Austria. GST-mono-ubiquitin was generated by insertion of the human ubiquitin coding sequence into pGEX-4T-1 vector.

GST-tagged proteins were expressed in *E. coli* Rosetta (DE3) pLysS cells. Cells were grown in LB medium and induced at $OD_{600} \approx 0.8–1$ for 4 hr at 37°C with 1 mM IPTG.

Harvested cells were resuspended in a buffer containing 50 mM HEPES at pH 7.5, 300 mM NaCl, 2 mM $MgCl_2$, 2 mM β-mercaptoethanol, complete protease inhibitor (Roche) and DNase I and lysed by freeze–thaw followed by sonication. Lysates were cleared by ultracentrifugation (140,000 *g* for 30 min at 4°C in a Beckman Ti45 rotor) and supernatants were applied to glutathione beads (GE Healthcare) for 1 hr at 4°C. Beads were washed five times with 50 mM HEPES, 300 mM NaCl, 1 mM DTT. GST-tagged proteins were eluted with 20 mM reduced L-glutathione in 50 mM HEPES pH 7.5, 300 mM NaCl, 1 mM DTT buffer for 1 hr at room temperature. The supernatant was concentrated and applied to a Superdex 75 column (16/600, GE Healthcare) previously equilibrated with 25 mM HEPES at pH 7.5, 150 mM NaCl, 1 mM DTT. Fractions containing purified proteins were pooled, concentrated, frozen in liquid nitrogen, and stored at -80°C.

eGFP-LC3B and eGFP-GABARAP were obtained by insertion of human LC3B and GABARAP cDNAs into pEGFP-C1. Fusion proteins were subsequently cloned into pETDuet-1 for bacterial expression. The last five amino acids of LC3B coding sequence and the last amino acid of GABARAP were deleted to mimic Atg4 cleavage. A 6xHis-tag was added C-terminally to recruit the protein to membranes preserving their physiological orientation.

To generate a monomeric meGFP-ubiquitin construct, mono-ubiquitin was cloned into pmeGFP-C3 vector, which encodes a monomeric enhanced GFP (*Zacharias et al., 2002*), N-terminally of the cloning site. The fusion protein was subsequently subcloned into the pETDuet-1 vector. A TEV site was added with the forward primer to generate 6xHis-TEV-meGFP-ubiquitin. Blue fluorescently tagged ubiquitin was generated inserting mTAG-BFP into pETDuet1 to generate 6xHis-TEV-BFP followed by insertion of ubiquitin.

Fluorescently tagged LC3B, ubiquitin and GABARAP were expressed in *E. coli* Rosetta (DE3) pLysS cells. Cells were induced at an $OD_{600}$ of 0.5 for 16 hr at 18°C with 0.1 mM IPTG. Proteins were purified on Ni-NTA columns as described above. Eluted eGFP-LC3B-6xHis and eGFP-GABARAP-6xHis were concentrated and directly applied to Superdex 75 column (16/60, GE Healthcare). 6xHis-meGFP-ubiquitin and 6xHis-BFP-ubiquitin were subjected to overnight TEV cleavage prior to SEC. Proteins were eluted in 25 mM HEPES pH 7.5, 150 mM NaCl, 1 mM DTT buffer, concentrated, frozen in liquid nitrogen and stored at -80°C.

LC3B-6xHis was generated by insertion of human LC3B into pETDuet 1. The last five amino acids of the coding sequence were deleted to mimic Atg4 cleavage. A 6xHis-tag was added C-terminally to recruit the protein to membranes preserving the physiological orientation. The protein was expressed overnight at 18°C in *E. coli* Rosetta (DE3) pLysS cells in the presence of 0.1mM IPTG and subsequently purified via His-Trap and SEC using a Superdex 75 column (16/60, GE Healthcare).

Recombinant human tetra-ubiquitin (K48 and K63-linked) were purchased from Boston Biochem, Cambridge, MA, USA. The lyophilized powder was resuspended in SEC buffer (25 mM HEPES pH = 7, 150 mM NaCl, 1mM DTT) to a final concentration of 100 μM. Linear tetra-ubiquitin was generated from GST-tetra-ubiquitin by overnight thrombin cleavage at 4°C and subsequent purification via SEC (Superdex 75 16/60, GE Healthcare).

## Analytical SEC and SLS

For analytical SEC, 200 μg of the mCherry-p62 variants were applied to a Superose 6 column (10/300, GE Healthcare) or Superdex 200 column (10/300, GE Healthcare) and eluted with 25 mM HEPES pH 7.5, 500 mM NaCl, 1mM DTT. 25 μL of 0.5 mL fractions were run on a 4-–20% SDS-PAGE gel (Biorad, Hercules, CA, USA) and stained with Coomassie. SLS analysis was done with a Superdex 200 column (10/300, GE Healthcare). Online Multi-Angle Laser Light Scattering detection was performed with a MiniDawn Treos detector (Wyatt Technology, Santa Barbara, CA, USA) via a laser emitting at 690 nm and by refractive index measurement using a Shodex RI-101 (Shodex, Munich, Germany).

## p62 (co-) sedimentation assay

Sedimentation behavior of the different p62 variants was analyzed by ultracentrifugation of 1 μM p62 solutions at 150,000 *g* for 1 hr 30 min at 4°C. Supernatant and pellet fractions were compared to the protein input by SDS-PAGE followed by Coomassie staining.

For p62 co-sedimentation assays with M1, K48 and K63-linked tetra-ubiquitin chains or with LC3B, proteins were incubated at a p62:Ub or p62:LC3B molar ratio of 1:4 (unless otherwise stated), for 1 hr on ice, before ultracentrifugation. The amount of p62 in the supernatant and pellet fractions was measured by gel densitometry using the ImageJ software.

## Protein cross-linking to beads

meGFP-ubiquitin, BFP-ubiquitin, and 6xHis-eGFP were cross-linked to carboxylated latex beads (4% w/v, Invitrogen) with a diameter of 2 µm. Next, 50 µL of a 100 µM protein solution in 50 mM MES pH 6.0 were added to 25 µL beads suspension, diluted 1:1 in 50 mM MES pH 6, and incubated at room temperature for 15 min. For direct cross-linking of p62 to beads, 50 µL of 50 µM mCherry-p62 delta PB1 and mCherry-p62 delta PB1 LIR were used.

0.8 mg of 1-ethyl-3-(3-dimethylaminopropyl)carbodiimide were added to the mix and further incubated for 2 hr at room temperature. The reaction was quenched by the addition of glycine to a final concentration of 100 mM, followed by incubation for 30 min at room temperature. Beads were washed three times with phosphate-buffered saline (PBS) and resuspended in 100 µL of buffer containing 1% bovine serum albumin (BSA) in 15 mM HEPES pH 7.5 and 135 mM NaCl. Cross-linked beads were stored at 4°C.

For HeLa cell transfection, beads were incubated with 0.25% BSA in PBS at room temperature for 15 min. After two washes with PBS, beads were stored in PBS at 4°C (~10 µg/µL).

For M1- K48- or K63-linked tetra-ubiquitin chains, 29.4 µL of a 100 µM protein solution were cross-linked to 29.4 µL of 2% 2 mm latex beads in 50 mM MES pH 6.0. For mock cross-link, the same amount of SEC buffer was used. Beads were finally resuspended in 58.8 µL of 1% BSA, 15 mM HEPES at pH 7.5, 135 mM NaCl buffer, and stored at 4°C.

## GUV formation

Lipids were purchased from Avanti Polar Lipids, Alabaster, AL, USA. GUVs were formed by electro-formation at 30°C as previously described (*Romanov et al., 2012*) For protein recruitment, a mixture of 95% 1-palmitoyl-2-oleoyl-sn-glycero-3-phosphocholine (POPC) and 5% 1,2-dioleoyl-*sn*-glycero-3-[(*N*-(5-amino-1-carboxypentyl)iminodiacetic acid)succinyl] (nickel salt) (DGS-Ni-NTA) was used (molar ratio). For beads engulfment experiments with p62 cross-linked to beads or GST-4xUb-coated beads, a mixture of 46% POPC, 46% 1,2-dioleoyl-*sn*-glycero-3-phosphocholine (DOPC), 5% DGS-Ni-NTA, 3% oregon-green phosphatidylethanolamine (Oregon-green PE) was used (molar ratio). For cross-linked ubiquitin chains experiment a mixture of 90% DOPC, 5% DGS-NiNTA, 5% Oregon-green DHPE was used.

## Protein recruitment to GUVs and membrane bending

Electroformed GUVs were diluted 1:2 to 1:3 in GUV buffer (15 mM HEPES pH 7.5, 135 mM NaCl, 1 mM DTT). The DOPC-containing GUVs were not diluted. eGFP-LC3B-6xHis was added at a final concentration of 400 nM. The mixture was incubated for at least 30 min at room temperature. mCherry-p62 variants were added to final concentration of 100 nM. Proteins were incubated for at least 30 min before imaging.

For membrane bending experiments with cross-linked p62 on beads, LC3B-6xHis was added to GUVs at a final concentration of 200 nM and incubated for 30 min at room temperature. Beads were spun at 2000 rpm for 30 s to precipitate aggregates, then 10 µL of the supernatant was added to the GUVs and incubated for 30 min before imaging. For experiments with GST-linear tetra-ubiquitin-coated beads, 2 µm glutathione beads (Sperotech, Lake Forest, IL, USA) were used. In total, 100 µL of beads slurry were spun at 160 *g* for 5 min and washed once with GUV buffer 0.25% BSA. Then, 75 µg of GST-linear tetra-ubiquitin were recruited to beads for 30 min at 4°C in 150 µL GUV buffer. Beads were then washed once with GUV Buffer 0.25% BSA, divided into three aliquots, and each sample was incubated with 50 µL of 2 µM mCherry-p62 solution for 1 hr at 4°C. Beads were washed once and resuspended in 100 µL of GUV buffer. 10 µL of beads suspension were added to GUVs and incubated for 30 min before imaging.

For experiments with ubiquitin chains cross-linked to beads, LC3B-6xHis was recruited to GUVs at 100 nM final concentration. mCherry-p62 wild type was recruited to beads at 500 nM final concentration for 1 hr at 4°C. At the end of the incubation, beads were sonicated for 5 s on ice, spun 2 min

at 3500 g, half of the supernatant was removed and beads were resuspended in the remaining 25 μL volume. Then, 5 μL were added to GUVs to a total reaction volume of 45 μL.

## Protein recruitment to ubiquitin-coated beads

Before each use cross-linked meGFP-ubiquitin, BFP-ubiquitin, and 6xHis-eGFP beads were resuspended by vortexing, an aliquot was diluted 1:100 and sonicated for 15 min in ice water. Diluted beads were incubated with mCherry-p62 variants at a final protein concentration of 50 nM for at least 20 min at room temperature before imaging at a confocal spinning disk microscope (Visitron, Puchheim, Germany).

M1- K48- or K63-linked tetra-ubiquitin cross-linked beads were diluted 1:50 in SEC buffer and 2.5 μL were added to 22.5 μL of a 0.1 μM mCherry-p62 solution. Samples were incubated at room temperature for at least 20 min prior to imaging.

## GST pull-down and microscopy-based assays using glutathione beads

20 μl of Sepharose 4B glutathione beads (GE Healthcare) were used. For each reactionbeads were equilibrated by three washes with NETN-E buffer (*Pankiv et al., 2007*). GST-tagged proteins were recruited to beads at 1 μM final concentration for 30 min at 4°C. Beads were washed once in NETN-E buffer and mCherry-p62 variants were added at 100 nM in a total volume of 55 μL. Beads were incubated for 1 hr at 4°C on an orbital shaker, washed twice with NETN-E buffer, and resuspended in 20 μL of 2× Laemmli loading buffer.

Protein recruitment assays to glutathione beads were performed in SEC buffer (25 mM HEPES pH 7.5, 150 mM NaCl, 1 mM DTT). Beads were incubated with 1.5 μg GST, GST-LC3B, GST-mono-, di-, or tetra-ubiquitin per μL beads for 30 min at 4°C, washed once and incubated with 2.5 μL per μL beads of 2 μM mCherry-p62 solution for 1 hr at 4°C. To measure p62 association to GST-LC3B-coated beads, imaging started immediately (1 s) after the addition of mCherry-p62 solution to the beads. For co-recruitment of LC3B and p62 on ubiquitin-coated beads, mCherry-p62 and GFP-LC3B were incubated together with the beads at 2 μM and 1 μM, respectively. For GST-LC3B and GST-mono-ubiquitin titration, the total amount of protein recruited on beads was kept constant, but decreasing amounts of GST-LC3B and GST-mono-ubiquitin were mixed with increasing amounts of GST. For GST-LC3B, beads were incubated with 1.5 μg of total GST/GST-LC3B mixture per μL beads and for GST-mono-ubiquitin with 2 μg/μL beads.

At the end of incubation 7.5 μL beads were diluted into 50 μL of p62 protein solution (for steady-state imaging and FRAP) or empty buffer (for decay assays) and imaged within a few minutes from dilution using a spinning disk microscope (Visitron).

## Protein recruitment to RFP-Trap beads

RFP-Trap agarose beads were used (ChromoTek, Martinsried, Germany). 20 μL beads were incubated with 50 μL of 2 μM mCherry-p62 and GFP-LC3B solution for 1 hr at RT. Beads were diluted in the same protein solution for steady-state imaging or in empty buffer for decay assays.

## FRAP

The FRAP experiments were performed with GST-LC3B or GST-ubiquitin-coated beads, incubated with mCherry-p62 variants according to the GST pull-down assay protocol. Defined areas on the beads' surface were photo-bleached using a 405 nm laser at 100% laser power for 50 ms per pixel and a 10 pixel-wide laser beam. Beads were imaged before and after photo-bleaching using a spinning disk microscope (Visitron). Fluorescence recovery was recorded for the indicated time. Recovery half-times were calculated by fitting the curves to a mono-exponential equation with plateau set at 100%.

## Cell culture

HeLa human epithelial cells (CCL-2, ATCC) were cultured in Dulbecco's modified Eagle medium (DMEM) high glucose, GlutaMAX, pyruvate (Gibco, Waltham, MA, USA) supplemented with 10% heat-inactivated fetal bovine serum (FBS, Sigma, St. Louis, MO, USA) and 100 units/mL penicillin and 100 μg/mL streptomycin (Gibco) at 37°C and 5% $CO_2$. Cells were used from passages 2 to 20.

## Transient transfection of siRNA, plasmids, and beads

$1 \times 10^5$ HeLa cells were seeded (for IF on a glass coverslip) in a well of a 6-well plate on day 1. Transfection of siRNA against endogenous SQSTM1/p62 (sip62) or control siRNA (siControl) was performed on day 2, followed by (co-)transfection of appropriate DNA constructs on day 4. On day 5, the assay was performed.

In total, 50 pmol/well of ON-TARGETplus human SQSTM1/p62 siRNA (J-010230-05, Dharmacon, Buckinghamshire, UK) or ON-TARGETplus nontargeting pool (D-001810-10, Dharmacon) together with 2.5 µL Lipofectamine RNAiMax (Invitrogen, Waltham, MA, USA) was incubated with serum-free medium for 20 min at room temperature and added to HeLa cells in 2 mL culture medium.

siRNA-resistant p62 variants in pmCherry-C1 with silent mutations (forward nucleotide sequence: ORF 970GAgCAaATGGAaTCcGAc987), full-length LC3B, full-length GABARAP, and/or mono-ubiquitin in pEGFP-C1 were used for transfection. 0.75 µg DNA for single transfection, 1.0 µg total DNA for co-transfection, or 2 µg BFP-ubiquitin cross-linked latex beads (2 µm) were pre-incubated with FuGene6 (Promega, Madison, WI, USA) in a 1 µg:3 µL ratio (DNA or beads:Fugene6) in serum-free medium. After 20 min at room temperature this mix was added to cells supplemented with fresh 2 mL (DNA) or 1 mL (beads) culture medium per well. Samples with beads were centrifuged at 175 $g$ for 5 min at room temperature to settle down beads, followed by three washes with PBS after 1 hr and an additional centrifugation step. After another 3 hr, cells were washed once with PBS and fixed with 3% paraformaldehyde for 20 min at room temperature.

## GFP-TRAP affinity purification

siRNA and/or DNA transfected HeLa cells were washed once with PBS and lysed in 100 µL/well lysis buffer containing 20 mM Tris pH 8.0, 10% glycerol, 135 mM NaCl, 0.5% NP-40, and protease inhibitors (Complete, EDTA-free, Roche) for 15 min on ice. After scraping the cells off, lysates were centrifuged at 16,100 $g$ for 5 min at 4°C to remove cell debris. In total, 150 µL of the lysis buffer without NP-40 or protease inhibitors (wash buffer) was added to the supernatant and this input was incubated with a mix of 2 µL GFP-TRAP_A beads (ChromoTek) and 8 µL empty Sepharose 4B beads (Sigma), equilibrated in wash buffer, for 1 hr at 4°C. After washing the beads 3× with wash buffer, beads were taken up in Laemmli loading buffer, boiled for 10 min at 95°C, and loaded on a SDS-PAGE. Proteins were detected by western blotting.

## Antibodies

The mouse monoclonal anti-GST antibody (clone 2H3-D10, diluted 1:1000) is available from Sigma. The mouse anti-GFP antibody (clones 7.1 and 13.1, diluted 1:1000 to 1:5000) was purchased from Roche (order number 11814460001). The monoclonal anti-LC3B (clone 2G6, diluted 1:500 for immunoblotting or 1:100 for immunofluorescence) is available from NanoTools, Teningen, Germany. The mouse monoclonal anti-p62 antibody (diluted 1:1000 to 1:5000 for immunoblotting or 1:100 for immunofluorescence) was purchased from BD Transduction Laboratories, Franklin Lakes, NJ, USA (order number 610832). The polyclonal BacTrace goat anti-*Salmonella* CSA-1 antibody (diluted 1:200) was purchased from KPL, Gaithersburg, MD, USA (order number 01-91-99). The mouse anti-GAPDH (clone GAPDH-71.1, diluted 1:50,000) is available from Sigma. The rabbit anti-ubiquitin serum is available from Sigma. Secondary antibodies for immunofluorescence were Alexa Fluor 488 or 546-conjugated goat anti-mouse IgG (diluted 1:1000) from Invitrogen, Alexa Fluor 647-conjugated goat anti-rabbit IgG (diluted 1:500) and Cy™5-conjugated AffiniPure donkey anti-goat IgG (diluted 1:400) from Jackson ImmunoResearch Laboratories, West Grove, PA, USA and Alexa Fluor 405-conjugated donkey anti goat IgG (diluted 1:200) from Abcam, Cambridge, UK.

## Immunofluorescence

After paraformaldehyde fixation, cells were washed 4× with PBS and permeabilized with 0.1% saponin (AppliChem, Darmstadt, Germany) in PBS (washing buffer) for 10 min at room temperature. After blocking with 5% BSA in washing buffer for 1 hr, cells were incubated with the primary antibody for 1 hr followed by three washes and the secondary antibody for another hour at room temperature. After three washes cells were mounted with Dapi fluoromount-G (SouthernBiotech, Birmingham, AL, USA) and observed on confocal LSM 710 or LSM 700 (Zeiss, Jena, Germany) microscopes. To

distinguish internal from external beads or *Salmonella*, IF was first performed on non-permeabilized cells (without saponin) followed by IF on permeabilized cells and slides were mounted with ProLong Gold Antifade, Invitrogen. For endogenous LC3B detection, cells were fixed in -20˚C cold methanol for 5 min at -20˚C.

## Bacterial infection and gentamicin protection assay

*S. typhimurium* LT2 wild type was cultured in LB medium with additional 300 mM NaCl overnight at 37˚C at 200 rpm. The next day the culture was diluted to $OD_{600}$ = ~0.2 and grown to $OD_{600}$ = ~0.9. siRNA and DNA pre-transfected HeLa cells were washed 2× with PBS and transferred into culture medium without antibiotics at least 2 hr prior to infection. HeLa cells of one well of a 6-well plate were counted and the required inoculum of *Salmonella* was determined for MOI 50 assuming that $OD_{600}$ = 0.9 corresponds to $1 \times 10^9$ *Salmonella*/mL. The inoculum was added to 1 mL culture medium without antibiotics/well and spun down at 300 *g* for 5 min at room temperature to synchronize infection. After half an hour of incubation at 37˚C and 5% $CO_2$, HeLa cells were washed 3× with PBS and DMEM with 10% FBS and 100 µg/mL gentamicin (Sigma) was added (time point 0 hr post infection). After 1 hr, HeLa cells were washed 2× with PBS and fixed with 3% paraformaldehyde in PBS for 20 min at room temperature. *Salmonella* were stained with 4′,6-diamidino-2-phenylindole (DAPI) or the anti-*Salmonella* antibody and imaged using confocal LSM 710 (Zeiss) or LSM 700 (Zeiss) microscopes.

## Quantifications and statistical methods

For quantification of protein recruitment to GUVs or beads, one line was drawn across each GUV/bead so that contact points between GUVs/beads as well as protein aggregates would be excluded. The average brightness of an empty portion of each picture was considered as the background for that picture (*Bkg*). For each line drawn, the protein binding intensity was calculated as the result of the difference (*Max - Bkg*), where *Max* denotes the maximal brightness across the line. Where the intensity of two fluorescent proteins was measured, the recruitment of each individual protein was calculated as described. Each bead/GUV was quantified in the same position for both proteins. The recruitment ratio of the two proteins was then calculated dividing the recruitment of the prey protein by the recruitment of the bait protein at the same position for each GUV/bead.

For fluorescence decay experiments, at least three fields were acquired per each individual sample. Every field was imaged as a Z-stack spanning all the beads contained in it. Time points were taken every 1.25, 2.5, or 5 min over a total time of 90 min. For quantification, Z-stacks corresponding to the shown time points were projected in single pictures as maximal Z projection (ImageJ); pictures were assembled in time-lapse stacks and the same positions in every slice were quantified as described. Fluorescence intensities of each bead at every time point were related to respective initial intensities at time point 0 as 100%.

FRAP curves were quantified measuring maximal fluorescence intensity at the bead's rim in the bleached region. The pre-bleaching value was set to 100% and first the post-bleaching time point to 0%. Recovery (*r*) at any following time point ($i_x$) was calculated as a fraction of pre-bleach minus post-bleach delta, that is, $r_{ix} = (i_x – i_{0\%})/(i_{100\%} – i_0\%)*100$.

For quantification of membrane bending, only contact points between beads and GUVs were considered. Bending was scored when the membrane was seen deflected or interrupted in correspondence of the bead.

Recruitment of mCherry-p62 variants and/or eGFP-LC3Bto beads in HeLa cells was determined by considering only internalized beads (negative for extracellular anti-ubiquitin antibody signal, but positive for BFP-ubiquitin) in mCherry-p62 and eGFP-LC3B co-expressing cells, with endogenous p62 knockdown. *Puncta* to ring structures showing the proteins directly at the beads were counted as positive localization to beads.

For quantification of protein recruitment at internalized bacteria, values are expressed as % of positive bacteria for the indicated protein over the total internal bacteria detected.

Unless differently stated, for all statistical analyses a two-tailed, unpaired Student's *t*-tests were performed.

## Acknowledgements

We thank Bettina Zens for assistance during protein expression. The research leading to these results has received funding from the European Research Council under the European Community's Seventh Framework Programme (FP7/2007-2013)/ERC grant agreement No. 260304, from the FWF Austrian Science Fund (grant number P25546-B20) and by the EMBO Young Investigator Program to SM. We also acknowledge funding by the Uni:docs program of the University of Vienna to GZ and funding by OEAW DOC program to CA.

## Additional information

### Funding

| Funder | Grant reference number | Author |
|---|---|---|
| European Research Council | ERC-StG 260304 | Sascha Martens |
| Austrian Science Fund | P 25546-B20 | Sascha Martens |
| EMBO | EMBO YIP Program | Sascha Martens |
| Universität Wien | Uni:docs fellowship | Gabriele Zaffagnini |
| Austrian Academy of Sciences | DOC fellowship | Christine Abert |

The funders had no role in study design, data collection and interpretation, or the decision to submit the work for publication.

### Author contributions

BW, Conception and design, Acquisition of data, Analysis and interpretation of data, Contributed unpublished essential data or reagents; GZ, Conception and design, Acquisition of data, Analysis and interpretation of data, Drafting or revising the article; DF, ET, CA, Conception and design, Acquisition of data, Analysis and interpretation of data; JR, Acquisition of data, Analysis and interpretation of data; SM, Conception and design, Analysis and interpretation of data, Drafting or revising the article, Contributed unpublished essential data or reagents

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
