## [Decision Letter]

Thank you for submitting your work entitled "Oligomerization of p62 allows for
selection of ubiquitinated cargo and isolation membrane during selective autophagy"
for peer review at *eLife*. Your submission has been favorably evaluated
by Randy Schekman (Senior editor and Reviewing editor) and three reviewers.

The reviewers have discussed the reviews with one another and the Reviewing editor has
drafted this decision to help you prepare a revised submission.

Summary:

This work describes a biochemical approach to investigate the relationship between p62
oligomerization and cargo binding. Contrary to Atg19, a substrate adaptor with multiple
LIR motifs, p62 contains only a single LIR-motif and a single UBA-domain. This study
suggests that oligomerization provides p62 with multiple LIR-motifs and UBA domains
within a single structure, probably related to the large helical structures previously
described by cryo-EM. Oligomerization decreases the off-rate of cargo from the adaptor,
thus stabilizing the interaction of adaptor and cargo sufficiently for membrane
formation around the cargo. In addition, the simultaneous binding of p62 to ubiquitin
and LC3 allows for bending of the autophagosomal membrane around cargo, at least in the
context of GUVs. There is also some correlation between in vitro data and recognition of
ubiquitylated Salmonella by p62.

Overall, there are several interesting observations in this manuscript, that paired with
careful biochemical characterization make it an interesting study for publication in
*eLife*.

Essential revisions:

1) My major concern with this study is the ubiquitin aspect: cargo for p62 is usually
heavily ubiquitylated, with either long chains or more complex ubiquitin structures
being observed most frequently. However, this study is mostly done in the context of
mono- or di-ubiquitin, and linkage-specificity of ubiquitin conjugates is not taken into
account. It is important to address both length and linkage of the ubiquitin conjugates
as previous work had suggested that long ubiquitin chains (octamers connected through
K63) can disrupt the oligomeric structure of p62. Thus, some of these assays need to be
repeated with defined K63- or K48-linked chains of at least four ubiquitin molecules
tested for their interaction with p62 and their ability to induce membrane bending in
conjunction with p62.

2) The assay in Figure 5 is nice. Are all
components required for bending the membrane or that an interaction between the beads
and the membrane could be sufficient to drive the curvature? The authors may need to
demonstrate a physiological circumstance which requires ubiquitin, oligomerized p62 and
LC3 to bend the membrane. Could the p62-coated beads bend the LC3-coated membrane? In
Figure 5, the data shown do not support the
statistical significance of the claim that "The oligomerization deficient mutants
of p62 showed reduced ability to mediate membrane bending (Figure 5)." The authors show that PB1-mediated
oligomerization is important for avid binding to Ub, but their results do not support
the idea it is a direct driving force for membrane bending.

3) Figure 6 do not appear to show significant
effects. In 6G in particular, the effects are very small. This is probably due to
adaptor redundancy. The authors speculate about other adaptors that might be involved.
On balance the data in Figure 6 neither
support nor oppose the authors' hypothesis.

4) Figure 7. While the histogram shows what
looks like an impressive reduction for the PB1 mutants, I'm confused about how
colocalization was scored. By eye it looks like plenty of GFP is colocalized with all of
the mutants in Figure 7.

5) The advance of the study is to illustrate the enhanced association of p62 with the
LC3 and ubiquitin by oligomerization. However, the process is insufficiently
characterized. The reviewer suggest calculating a binding curve between the lipid
anchored LC3 and different concentrations of different p62 variants. Further, the
binding kinetics in addition to the dissociation kinetics indicated in Figure 3 may also be helpful. If oligomerization is
just simply increasing the affinity to LC3 and ubiquitin binding, could an engineered
p62 with multiple LIFs fulfill similar purpose?

6) Oligomerization decreases the diffusion rate of p62 in the cytosol which may be a
negative contributor to its dynamics and efficiency to recognize cargo. Therefore, it is
possible that the extent of p62 oligomerization is low but enhanced by cargo binding or
association with LC3. Could the author examine the effect of ubiquitin and LC3 on the
extent of p62 oligomerization?

7) The author may consider revising the conclusion in the subsection "Oligomerization of p62 renders binding to LC3B-coated surfaces irreversible". The authors indicated the "concentrated p62". Here it is
difficult to conclude from Figure 3 that these
soluble LC3 is not concentrated, as once they associate with the oligomerized p62
concentrated on the beads, they may become concentrated too. The reviewer suggests
modifying "concentrated" to "membrane associated". Otherwise, it is
necessary to quantify the fluorescence intensity of the LC3 on the lipid surface as well
as those associated the p62-coated beads to make a comparison. In addition, the authors
may consider titrating the amount of LC3 on the membrane and quantifying the
dissociation rate of oligomerized p62 from the membranes coated with different
concentrations of LC3 proteins.

---

## [Author Response]

*1) My major concern with this study is the ubiquitin aspect: cargo for p62 is
usually heavily ubiquitylated, with either long chains or more complex ubiquitin
structures being observed most frequently. However, this study is mostly done in the
context of mono- or di-ubiquitin, and linkage-specificity of ubiquitin conjugates is
not taken into account. It is important to address both length and linkage of the
ubiquitin conjugates as previous work had suggested that long ubiquitin chains
(octamers connected through K63) can disrupt the oligomeric structure of p62. Thus,
some of these assays need to be repeated with defined K63- or K48-linked chains of at
least four ubiquitin molecules tested for their interaction with p62 and their
ability to induce membrane bending in conjunction with p62.*

Thank you very much for this important point. We have now tested the interaction of wild
type p62 and the non-oligomeric delta PB1 mutant with beads cross-linked to linear
tetra-ubiquitin, K48-linked tetra-ubiquitin and K63-linked tetra-ubiquitin (Figure 4). Interestingly, while the delta-PB1
mutant bound equally to all three chain types, the oligomeric wild type protein showed
increased binding to linear ubiquitin and to some extent to K63-linked ubiquitin chains
but not to K48-linked ubiquitin. When we tested the oligomer disruption activity of the
three chain types we found that K48-linked ubiquitin chains had the strongest effect on
oligomerization (Figure 4). Thus it appears that
K48-linked ubiquitin chains may not be the preferred target for oligomeric p62. We
further tested the effect of ubiquitin chain length on p62 binding using mono-ubiquitin
and linear di- and tetra-ubiquitin (Figure 4).
Both, wild type and p62 delta PB1 bound stronger to longer chain types but the positive
effect of the increasing chain length was more pronounced for the oligomeric wild type
protein. This was despite the fact that at high concentrations linear ubiquitin chains
have a considerable disruptive effect on p62 oligomerization (Figure 4). We then went on to test the dissociation of p62 from
mono-ubiquitin and linear tetra-ubiquitin using FRAP (Figure 4) and found that p62 is much more tightly bound (i.e. has a lower
off rate) to tetra-ubiquitin chains than to mono-ubiquitin. Increasing the density of
mono-ubiquitin on the beads also resulted in reduced off-rates of p62 from the beads
(Figure 4). Thus a major factor for the avid
binding of oligomeric p62 to ubiquitinated structures is the density of ubiquitin rather
than the existence of chains, although chains could still have an additive positive
effect. We also tested the ability of wild type p62 to bend the membrane of GUVs around
beads coated with linear tetra-ubiquitin, K48-linked tetra-ubiquitin and K63-linked
tetra-ubiquitin (Figure 5). We found that p62
mediated membrane bending for all three chain types. Consistent with weaker interaction
of p62 with K48-linked chains bending was less efficient for this chain type.

*2) The assay in*Figure 5*is nice. Are all components required for bending the membrane or
that an interaction between the beads and the membrane could be sufficient to drive
the curvature? The authors may need to demonstrate a physiological circumstance which
requires ubiquitin, oligomerized p62 and LC3 to bend the membrane. Could the
p62-coated beads bend the LC3-coated membrane? In*
Figure 5*, the data shown do not support
the statistical significance of the claim that "The oligomerization deficient
mutants of p62 showed reduced ability to mediate membrane bending* (Figure 5)*." The authors show that
PB1-mediated oligomerization is important for avid binding to Ub, but their results
do not support the idea it is a direct driving force for membrane bending.*

We have now tested beads directly cross-linked to p62. In order to prevent massive
crosslinking of beads by oligomeric p62 we have used delta PB1 and the corresponding LIR
mutant for this experiment. We found that cross-linked p62 delta PB1 efficiently
mediated membrane bending (Figure 5) showing
that the presence of ubiquitin per se is not essential for the membrane bending activity
of p62. In addition, this experiment shows that oligomerization is also not essential
for membrane bending (although it could of course have an additive positive effect).
Thus oligomerization may primarily mediate the accumulation of p62 at the ubiquitinated
cargo, which in turn clusters the LIR motifs allowing the bending of the LC3B-coated
membrane. We thank the reviewers for suggesting this insightful experiment.

Regarding the experiment shown in Figure 5 of
the previous version of the manuscript (now replaced with Figure 5) in which we showed a comparison of the membrane-bending
activities of the p62 variants it is clear that it had some conceptual difficulties.
Thus wild type p62 binds very strongly to the membrane via its interaction with LC3B
(Figure 2, Figure 3) and the beads via its interaction with ubiquitin (Figure 4). This means that the protein will cover both surfaces and
due to the practically irreversible nature of these interactions the interaction of the
beads with the membrane are mediated by p62-p62 interactions. In contrast, the
oligomerization deficient mutants are less tightly bound to the membrane and the beads
and thus are likely to exchange and therefore to be able to mediate simultaneous
interaction with LC3B and ubiquitin. We came to the conclusion that we are not comparing
the same phenomenon for the different p62 variants. Therefore we have changed the
experimental setup by pre-incubating linear tetra-ubiquitin coupled beads with p62
followed by their addition to GUVs coated with LC3B (Figure 5). This experiment shows that p62 is able to bend the membrane
around the beads and that this activity is significantly reduced for the delta PB1
mutant.

*3)*
Figure 6*do not appear to show
significant effects. In 6G in particular, the effects are very small. This is
probably due to adaptor redundancy. The authors speculate about other adaptors that
might be involved. On balance the data in*
Figure 6
*neither support nor oppose the authors' hypothesis.*

We agree with the reviewer in that the effects of the oligomerization deficient mutants
of p62 on LC3B recruitment to the beads are very small. However, although the effects
are indeed very small they are significant. We have left the Figure as is but would be
happy to move Figure 6 to the Figure
Supplement or completely from the paper should the reviewers insist.

*4)*
Figure 7*. While the histogram shows what
looks like an impressive reduction for the PB1 mutants, I'm confused about how
colocalization was scored. By eye it looks like plenty of GFP is colocalized with all
of the mutants in*
Figure 7.

Figure 7 showed the degree of co-localization of
p62 and LC3B at the bacteria. While the degree of LC3B localization to the bacteria was
significantly reduced for oligomerization mutants (Figure 7) the level of co-localization of the p62 and LC3B at the bacteria
is mainly influenced by the reduced localization of the p62 oligomerization mutants to
the bacteria. We understand that this quantification can be misleading and have moved it
to the Figure Supplement ( Figure 7—figure supplement 2).

*5) The advance of the study is to illustrate the enhanced association of p62
with the LC3 and ubiquitin by oligomerization. However, the process is insufficiently
characterized. The reviewer suggest calculating a binding curve between the lipid
anchored LC3 and different concentrations of different p62 variants. Further, the
binding kinetics in addition to the dissociation kinetics indicated in*
Figure 3
*may also be helpful. If oligomerization is just simply increasing the affinity
to LC3 and ubiquitin binding, could an engineered p62 with multiple LIFRs fulfill
similar purpose?*

We thank the reviewer for suggesting these experiments. We have now tested the binding
of p62 to LC3 at different concentrations. Instead of GUVs we have used beads since they
also allowed us to conduct FRAP experiments without the complication of lateral
diffusion of LC3B and the LC3B-p62 complexes on the membrane. Figure 2 and Figure 2—figure supplement 3 show the association of wild type p62 as well as the delta PB1
and LIR mutants to LC3B coated beads at different concentrations. Figure 3—figure supplement 2shows the binding of wild type and
delta PB1 p62 to LC3B-coated beads overs time.

Figure 3 and Figure 3—figure supplement 3 show FRAP analyses of p62 bound to
different densities of LC3B on the beads. In combination these FRAP analyses show that
the avid binding of p62 to LC3B depends on the density of LC3B on the bead. At high
densities the interaction is practically irreversible while at lower densities the
effect of oligomerization is lost and the behaviors of oligomeric p62 is like that of
the delta PB1 mutant.

Figure 4 shows a FRAP analysis of the
interaction of p62 with different densities of ubiquitin on beads. The conclusion of
this experiment is that, analogous to the interaction with LC3B, the binding of wild
type p62 is more avid to higher ubiquitin densities.

As suggested by the reviewers we have also constructed a p62 version with multiple LC3B
interaction site (LIRs). In particular, we have introduced 3 additional LIR motifs next
to the endogenous LIR of p62 delta PB1. The FRAP analysis beautifully shows that this
4xLIR mutants binds much more avidly to LC3B coated beads than the single LIR containing
protein (Figure 3).

*6) Oligomerization decreases the diffusion rate of p62 in the cytosol which may
be a negative contributor to its dynamics and efficiency to recognize cargo.
Therefore, it is possible that the extent of p62 oligomerization is low but enhanced
by cargo binding or association with LC3. Could the author examine the effect of
ubiquitin and LC3 on the extent of p62 oligomerization?*

We have tested the effect of ubiquitin and LC3B binding on p62 oligomerization. To this
end we have employed a pelleting assay as described previously by the Sachse and
Johansen labs (Ciuffa et al., Cell Rep. 2015). The assay works well as the presence of
p62 in the pellet fraction correlates with ability of p62 to oligomerize (Figure 2). In agreement with Ciuffa et al. we
found no oligomerization disruption effect by LC3B (Figure 4—figure supplement 1). Also, in agreement with Ciuffa et al. we found
that ubiquitin disrupted p62 oligomers to a considerable extent. When we compared
different ubiquitin chains for their effect on oligomerization we found that linear,
K48-linked and K63-linked chains disrupted p62 oligomers (Figure 4). Of these K48-linked chains had the strongest disruptive
effect. However, oligomeric p62 still binds stronger to ubiquitin chains than to
mono-ubiquitin (Figure 4).

*7) The author may consider revising the conclusion in the subsection
"Oligomerization of p62 renders binding to LC3B-coated surfaces irreversible". The authors indicated the "concentrated p62". Here it
is difficult to conclude from*
Figure 3
*that these soluble LC3 is not concentrated, as once they associate with the
oligomerized p62 concentrated on the beads, they may become concentrated too. The
reviewer suggests modifying "concentrated" to "membrane
associated". Otherwise, it is necessary to quantify the fluorescence intensity
of the LC3 on the lipid surface as well as those associated the p62-coated beads to
make a comparison. In addition, the author may consider titrating the amount of LC3
on the membrane and quantifying the dissociation rate of oligomerized p62 from the
membranes coated with different concentrations of LC3 proteins.*

Thank you for pointing out this misleading statement. We now write "Taken together
these results suggest that oligomerization of p62 specifically promotes interaction with
surface-localized, clustered LC3B by drastically reducing the off-rate of p62 from
LC3B-coated surfaces (subsection "Oligomerization of p62 renders binding to
LC3B-coated surfaces irreversible, fourth paragraph).

As suggested by the reviewers we have tested the dissociation rate of p62 from beads
coated with different densities of LC3B (see response to point 5, Figure 3 and Figure 3—figure supplement 3 and Figure 3—figure supplement 4).